# Benchmarking Multivariate Time Series Anomaly Detection with Large-Scale Real-World Datasets

## Abstract

Time series anomaly detection is of significant importance in many real-world applications, including finance, healthcare, network security, industrial equipment, complex computing systems, space probes, etc. Most of them involve multi-sensor systems; thus how to perform multivariate time series anomaly detection (MTSAD) has attracted widespread attention. This broad attention has fueled extensive research endeavors aiming to innovate and develop methods and techniques to improve the efficiency and precision of anomaly detection on multivariate time series data, including classic machine learning methods and deep learning methods. However, how to evaluate the performance of all these methods is a challenging task. The first challenge lies in the limited public benchmark datasets for MTSAD, most of which are criticized from some perspectives. The second but related challenge is that the best metric for time series anomaly detection remains unclear, making the research in MTSAD hard to follow. In this paper, we advance the benchmarking of multivariate time series anomaly detection from datasets, evaluation metrics, and algorithm comparison. To the best of our knowledge, we have generated the largest real-world datasets for MTSAD from the Artificial Intelligence for IT Operations (AIOps) system for a real-time data warehouse in a leading cloud computing company. We review and compare popular evaluation metrics including recently proposed ones. To evaluate classic machine learning and recent deep learning methods fairly, we have performed extensive comparisons of these methods on various datasets. We believe our benchmarking and datasets can promote reproducible results and accelerate the progress of MTSAD research.

## 1 Introduction

Multivariate time series anomaly detection (MTSAD) (Blázquez-García et al., 2021) is critical in modern data analysis, with important applications in various fields such as finance, healthcare, industrial equipment, etc. However, although time series anomaly detection (TSAD) has been extensively researched, it is still challenging to identify the best model due to the varied datasets and metrics used in testing. This variability extends to different experimental settings, highlighting a significant limitation in the current TSAD research.

Currently, the multivariate time series benchmark datasets from real-world scenarios are lacking as it is usually expensive to label the anomalies for each time point. Recently, some new datasets have been proposed in the literature (Lai et al., 2021; Paparrizos et al., 2022b; Vincent Jacob & Tatbul, 2021). However, most of them are about univariate time series and synthetic data according to predefined abnormal patterns. It truly is of great importance to generate such datasets, but they cannot serve as the benchmark datasets for multivariate time series. More specially, multivariate time series are complex, and the anomalies have various patterns that are usually hard to synthesize. Thus, real-world datasets are significantly crucial for MTSAD. However, there are a limited number of public datasets for MTSAD now, and confusing labels and limited types of anomalies challenge the existing datasets.

The metrics for TSAD are another critical component of the evaluation. Unfortunately, there are many different metrics, such as F1-score with point adjustment, affiliation score, volume under the surface (VUS), etc. They focus on different detection objectives. However, we find that all of them still have limitations. An interesting question is, does there exist a perfect metric for all the MTSAD situations? We try to discuss this topic and show that metric designs should consider the applications as different situations may have different requirements. For example, in real-world industrial equipment, early warning is more important than postmortem diagnosis. Moreover, although precision and recall

are both important for evaluation, they do not have the same importance in a certain application. However, most of the existing metrics do not take these points into consideration.

With the advancement of deep learning, it also demonstrates superior performance in the field of TSAD. The evolution of deep learning in TSAD represents an exciting area of research, offering great potential for more accurate and efficient anomaly detection in various domains. However, this growth has also presented new challenges. Deep learning-based models are data-hungry and may overfit if not properly regularized, which can lead to false-positive detections. Furthermore, it is even harder to say which is the best deep model for TSAD as the settings may also not be consistent among the models. The "black-box" nature of deep learning models makes it even more difficult to interpret and compare different models. Thus, the fair comparison between different deep learning algorithms remains challenging. For example, a recent study (Lai et al., 2021) claims that classic algorithms outperform many recent deep learning approaches. However, as the state-of-the-art deep models are omitted, it may not be fair enough to make such a conclusion.

Considering all the above challenges in MTSAD, we aim to comprehensively benchmark MTSAD on both public datasets and real-world data collected from industrial application scenarios. **We summarize our main contributions as follows**:

1. **Largest Real-world Datasets for MTSAD**. To the best of our knowledge, we have collected the largest real-world datasets from the AIOps scenarios of a real-time data warehouse in a leading cloud computing company. We collected 256 instances over 120 days (one timestamp in each minute) and selected 48 instances of them, which are typical in different application scenarios. The dimension of instances ranges from 9 to 332, and the number of anomalies ranges from 2 to 9608. Totally our datasets contain 3611 time series with over 600 million points.
2. **Comprehensive Metrics Evaluation and Discussion with SOTA Models**. We discuss the designs of different metrics widely used nowadays and compare them with the same experiment settings. Furthermore, we evaluate a wide range of 14 models on both public datasets and our own datasets. These models include classic machine learning-based methods as well as major state-of-the-art deep learning-based methods.
3. **Accessible Benchmark and Datasets**. We make our benchmark and real-world datasets open-sourced[1] to promote reproducible results and accelerate the progress of MTSAD research.

## 2 LIMITATIONS OF EXISTING BENCHMARK

Although time series anomaly detection has been widely studied in recent years, it is still hard to tell which model is best. The main reason is that existing models are tested in different datasets with different metrics. Even the main experiment settings are diverse. What's worse, the public datasets and metrics widely used still have significant disadvantages. Another concern is that, although time series anomaly detection with deep learning has attracted great attention recently, they have not been compared systematically yet.

### 2.1 BENCHMARK

Existing TSAD benchmarks focus on different goals and solve only one or two challenges summarized above. UCR (Wu & Keogh, 2021) discussed the flaws of widely used public datasets and published a new dataset. However, in this univariate time series dataset, anomalies only happen at the tail of the sequence whose assumption is too strong. (Doshi et al., 2022) shows a surprising result that the widely used F1-score with point adjustment metric evaluates a rudimentary Random Guess method outperforms the state-of-the-art detectors. (Kim et al., 2022) notices that F1-score with point adjustment has the flaw of overestimating detection performance in theoretical and experimental views. Many new TSAD metrics have been proposed recently, but we will show even the latest one still has flaws. (Paparrizos et al., 2022b) focuses on univariate TSAD and proposes 13766 time series to enrich the existing limited public dataset. Most of the time series are synthetic and some of them are transformed from public classification datasets. (Vincent Jacob & Tatbul, 2021) considers a new explainable anomaly detection over time series and provides corresponding real-world AIOps data. In all the above works, deep learning-based TSAD models are not fully investigated.

### 2.2 DATASETS

Several public datasets are widely used by most of the existing models, including univariate time series datasets and multivariate time series datasets. For univariate time series anomaly detection tasks, Yahoo (N. Laptev & Billawala, 2015) and KPI (Competition, 2018) datasets are popular.

---

[1]https://anonymous.4open.science/r/AnomalyDetectionBenchmark-47F8/README.md

Table 1: A summary of frequently used public time series anomaly detection datasets and the proposed large-scale real-world AIOps dataset.

| Variable type | Dataset | #Curves/Dims | #Points | %Anomaly |
|---|---|---|---|---|
| Univariate | KPI | 58 | 343,528,954 | 2.26 |
| | Yahoo | 367 | 210,278,522 | 0.68 |
| Multivariate | SMAP | 27 | 15,791,220 | 12.8 |
| | MSL | 55 | 7,262,530 | 10.5 |
| | SMD | 38 | 53,839,350 | 4.2 |
| | SWaT | 51 | 48,190,869 | 12.1 |
| | PSM | 25 | 5,508,050 | 27.8 |
| | NIPS-TS-SWAN | 38 | 456,000 | 32.6 |
| | NIPS-TS-GECCO | 9 | 1,246,689 | 1.1 |
| | **AIOps** (this paper) | $9 \sim 332$ | 606,554,114 | $0.0012 \sim 5.72$ |

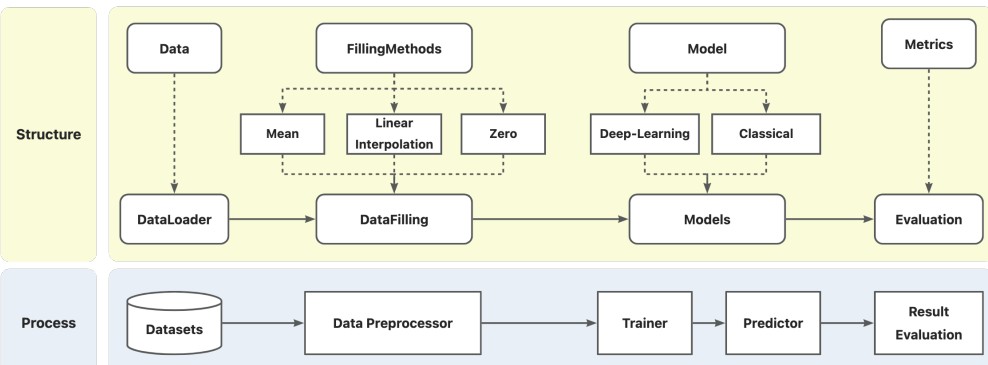

Figure 1: The overall benchmark framework for multivariate time series anomaly detection.

For multivariate time series anomaly detection tasks, NASA-MSL (Benecki et al., 2021), NASA-SMAP (Benecki et al., 2021), SMD (Su et al., 2019), SWaT (Mathur & Tippenhauer, 2016), and PSM (Abdulaal et al., 2021) are frequently used to evaluate the performance of designed models. The collection of existing real-world datasets is limited in the number of datasets and the size of each dataset. A summary of existing datasets is provided in Table 1. However, there are several flaws in existing public datasets: 1) limited collection of public real-world datasets; 2) confusing ground truth labels; and 3) limited types of time series anomalies. (Due to the space limit, more details about flaws in public datasets are left in the Appendix, Section A.) These flaws become even more severe in multivariate time series as discussed in the Appendix. Therefore, it is urgent to collect more real data for MTSAD.

## 3 MTSAD BENCHMARK DETAILS

### 3.1 BENCHMARK FRAMEWORK

As discussed above, there are serious flaws in TSAD, including flaws in public datasets, metrics, and model settings. The flaws become more challenging in MTSAD as the anomalies in multivariate time series are more complicated and harder to detect. Although anomalies in univariate time series can be classified as point-wise outliers and pattern-wise outliers with certain characteristics (Lai et al., 2021), it is unclear how to define them in multivariate time series.

The main difference between univariate and multivariate time series is the relation among different channels of multivariate time series. However, it is still difficult to define anomalies from this perspective, which partially explains why few synthetic multivariate time series anomaly detection datasets are proposed. Thus, we take multivariate time series data with anomalies in the real world to address challenges in datasets. Specifically, we will discuss the detailed settings of existing deep time series anomaly detection models and compare them in a fair setting with various metrics. The framework of our benchmark is shown in Figure 1. We take real-world AIOps data of a real-time data warehouse into consideration. As the lack of data usually happens, we provide three different filling NaN methods and focus more on the performance of deep learning models with diverse evaluation metrics.

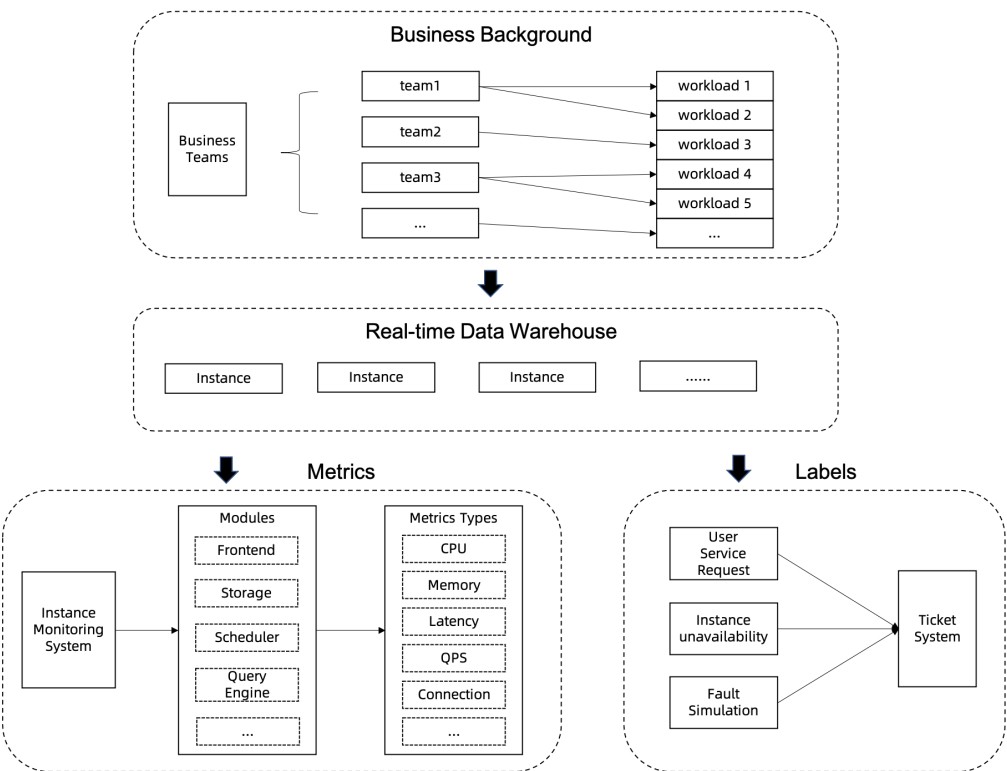

Figure 2: Illustration of the collection of AIOps datasets.

## 3.2 REAL-WORLD AIOPS DATASETS

### 3.2.1 BACKGROUND OF ANOMALY DETECTION FOR THE REAL-TIME DATA WAREHOUSE INSTANCES

The real-time data warehouse is a cloud-native service for hybrid serving and analytical processing (HSAP) developed by a leading cloud computing company. The system is designed for modern business analysis and can handle high-volume data ingestion, as well as the fusion of offline and online analysis. A database workload is a set of requests that have some common characteristics such as application, source of request, type of query, business priority, and performance objectives. The data warehouse supports hundreds of workloads belonging to different teams within the leading cloud computing company that support various business scenarios. Workloads in the data warehouse are hybrid and highly dynamic, usually subject to sudden bursts. Workloads can execute on a data warehouse instance, which represents a deployment of the data warehouse engine within a cloud environment and leverages various cloud resources, including computing, storage, and network capabilities. Given its high elasticity and scalability, the architecture of the data warehouse is complex and involves several key components, such as frontend nodes, storage managers, resource managers, and schedulers, as depicted in Figure 7 (in the Appendix). Inappropriate user operations, changes in user business, system module failures, and infrastructure breakdowns upon which the data warehouse relies, all have the potential to cause anomalous behavior within workloads executing on the instances. This can manifest in a range of different anomaly types, which are highly variable in nature.

Due to the complexity of the system, the large scale of the users and monitoring indicators, as well as the dynamicity of the workloads, it is hard for engineers to detect all the anomalies in time with human-defined alarm rules. In real-world applications, Mean Time to Resolve (MTTR) is a software term that measures the time period between a service being detected as "down" to a state of being "available" from a user's perspective, which is used by operations and development teams to support SLAs as shown in Figure 3. While the Mean Time to Detect (MTTD) represents the time period between the occurrence of a fault and the detection of an anomaly, it is also an essential component of the MTTR. The accuracy and efficiency of the MTTD can have a significant impact on the overall performance and reliability of the system. Therefore, anomaly detection algorithms play a vital role in ensuring SLAs by enabling engineers to detect anomalies in real-time data warehouse instances accurately and promptly. In most cases, these detected anomalies can be resolved by the self-healing module in the system, before they become noticeable to end-users.

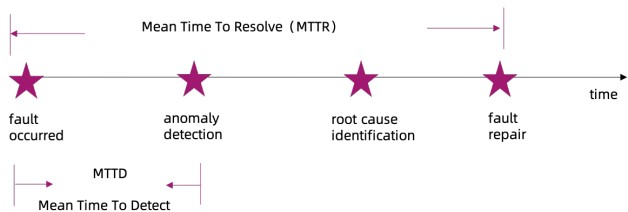

Figure 3: Illustration of MTTR and MTTD.

In addition to accuracy and real-time capability, an additional challenge in anomaly detection for real-time data warehouse instances pertains to the limitations of traditional univariable time series anomaly detection techniques, as they may not be sufficient to identify the true anomalies. Figure 8 (in the Appendix) illustrates a real-world example of an anomaly in a real-time data warehouse instance in which a five-pointed star indicates the timestamp of the anomaly. We observe that the anomaly cannot be identified using any one of the metrics presented in the figure. On the contrary, it is necessary to consider all three metrics collectively to make informed decisions and accurately identify anomalies. This example is just one among many real-world cases that which multiple metrics must be taken into consideration when dealing with the variety of anomalies that arise in a highly complex and large-scale system. Given the intricate nature of such systems, it is important to develop advanced MTSAD techniques that can effectively capture and analyze diverse data patterns from multiple sources.

### 3.2.2 DATA COLLECTION DETAILS

All the metrics and labels in our datasets are derived from real-world scenarios. Figure 2 illustrates the process of collecting metrics and labels to construct the datasets. All metrics were obtained from the real-time data warehouse monitoring system and cover a wide variety of metric types, including CPU usage, queries per second (QPS) and latency, which are related to many important modules within the real-time data warehouse. This comprehensive data collection strategy provides a multi-dimensional view of the performance of those instances, which is critical to accurately identify anomalies and maintain high levels of system reliability. In the multivariate datasets, the correctness of labels is vital, as they indicate whether an anomaly has occurred in a specific timestamp for a given instance. We obtain labels from the ticket system, which integrates three main sources of instance anomalies: user service requests, instance unavailability, and fault simulations. User service requests refer to tickets that are submitted directly by users, whereas instance unavailability is typically detected through existing monitoring tools or discovered by Site Reliability Engineers (SREs). Since the system is usually very stable, we augment the anomaly samples by conducting fault simulations. Fault simulation refers to a special type of anomaly, planned beforehand, which is introduced to the system to test its performance under extreme conditions. All records in the ticket system are subject to follow-up processing by engineers, who meticulously mark each ticket's start and end times. This rigorous approach ensures the accuracy of the labels in our datasets.

Based on the aforementioned introduction, several key characteristics of our datasets can be identified. Firstly, the datasets are large-scale and derived from real-world scenarios. Secondly, the datasets contain a wide range of anomalies, owing to the diverse sources of the ticket system and the variety of workloads. This poses a significant challenge to anomaly detection and necessitates the use of advanced machine learning algorithms and multivariate analytical approaches. Lastly, the labels are accurately annotated, ensuring the quality and reliability of the datasets.

### 3.2.3 STATISTIC CHARACTERISTIC OF THE AIOPS DATASETS

As shown in Section 1, our collected AIOps datasets include 48 instances with a time span of around 120 days. Each instance has a different number of dimensions, from 9 to 332. We show the information about some typical instances in AIOps datasets in Table 2.

### 3.3 EVALUATION METRICS

Besides datasets, evaluation metrics are also important for benchmarking. For the TSAD task, evaluation metrics are more crucial than other tasks (e.g., forecasting and classification) and anomaly detection in other fields. The main reason is that, for sequence-type data (time series is typical), there is no natural definition of 'sample'. A single point without contextual information in time series means nothing. However, contextual information is difficult to define, e.g., the context for abnormal time series with different lengths. Furthermore, in reality, different fields focus on different goals of anomaly detection. For example, in the medical scenario, recall is more important than precision, but

Table 2: Some instances information in AIOps datasets.

| Instance name | #Curves/Dims | #Timestamps | # Anomalies |
|---|---|---|---|
| instance0 | 21 | 167974 | 1664 |
| instance6 | 77 | 167974 | 300 |
| instance8 | 53 | 167974 | 9 |
| instance11 | 35 | 167974 | 46 |
| instance14 | 44 | 167974 | 1473 |
| instance15 | 16 | 167974 | 295 |
| instance23 | 56 | 167974 | 32 |
| instance35 | 37 | 167974 | 83 |
| instance38 | 27 | 167974 | 79 |
| instance39 | 121 | 167974 | 112 |
| instance44 | 39 | 167974 | 1271 |

in some industrial production fields, too many alarms may lead to a waste of human resources. The above challenges make the design of evaluation metrics difficult.

In this section, we will discuss most of the existing metrics in TSAD. All metrics can evaluate the algorithm's performance from some perspective, but also suffer from other limitations. Next, we will discuss more details of these evaluation metrics.

**F1-score with Point Adjustment**   This metric is proposed by Xu et.al (Xu et al., 2018; Audibert et al., 2020). It works as follows: if one anomaly point is correctly detected in the ground truth anomaly segment, all the points in such segment will be considered as correctly detected. Then F1-score is calculated with such adjusted predictions. F1 PA is designed with the alert that one detected anomaly shows errors in system sufficiently. However, this metric has a high possibility of overestimating the performance of models and does not consider the information of anomaly events. Actually, with the F1 PA metric, even random guess can gain SOTA performance (Kim et al., 2022; Doshi et al., 2022).

Besides the original F1 score with point adjustment, there are also several variants of F1 PA metric. For example, F1 PA%K (Kim et al., 2022) applies point adjustment only when the ratio of the number of correctly detected points is larger than the %K of anomaly length, where K is a threshold. It indeed helps relieve over-optimistic in F1 PA metric, but does not solve the problem of F1 PA. The gut issue of F1 PA is that it considers anomalies in point view. However, in time series field, it is hard to say it is reasonable to consider a single point as a sample. To gain information, a more natural way is to define something like anomaly events.

**Composite F1-score**   This metric takes event-wise anomaly into account (Garg et al., 2021) but still keeps the main design of point adjustment. In particular, it takes a point-wise precision with PA and an event-wise recall. The formalization is written as follows

$$Pr_t = \frac{TP_t}{TP_t + FP_t} \quad \text{and} \quad Rec_e = \frac{TP_e}{TP_e + FN_e} \tag{1}$$

where $TP_t$ and $FP_t$ are the number of TP and FP points respectively, $TP_e$ and $FN_e$ are the number of TP and FN events respectively. $TP_e$ is the number of true events for which at least one point is detected correctly. The other true events are counted under $FN_e$. This metric does not differentiate the locations of false positive events and over punish missing detection of single point events. Moreover, it is not very persuasive that precision and recall should be defined in different views. Actually, there are metrics taking the position of results into consideration, like NAB score (Lavin & Ahmad, 2015), SPD score (Doshi et al., 2022).

**Affiliation Score**   Affiliation (Huet et al., 2022) is a metric with an intuitive interpretation where both precision and recall are calculated based on the distance between ground truth and prediction events. Event distance is defined through point sets by Hausdorff distance (Dubuisson & Jain, 1994), and precision/recall is set by individual probability based on event distance normalized by affiliation zone. Affiliation is proven to be robust against adversary strategies. It is novel to measure event distance by Hausdorff distance and exquisite to draw individual probability into precision and recall. However, the affiliation zone has a huge influence on the final score. With little improvement in precision, the bigger size of the zone results in a higher score in a non-negligible degree. Moreover, all the prediction events in the zone contribute to the final score, even when they are false positives. Actually, if a prediction event is far from the ground truth, it should be punished. Another phenomenon from the affiliation zone is that it exhibits high tolerance for false positives but low tolerance for false negative points.

**VUS Metric** Besides the above metrics based on F1 score, there are also metrics based on receiver operator characteristic (ROC) curve and the area under the curve (AUC). The original ROC and AUC are based on point-wise detection. As discussed above, such point-wise type metrics introduce unavoidable shortcomings in range-based anomalies by mapping discrete labels into continuous data. That is why event-based F1-scores appear. Volume Under the Surface (VUS) metric (Paparrizos et al., 2022a) extends the AUC-based measures to account for range-based anomalies. The key designs are label transformation technique and volume under the surface metric. For label transformation, with a buffer length, the binary label is extended into a continuous value. Given buffer length $l$, the positions $s, e \in [0, |label|]$ are the beginning and end indexes of a labeled (range) anomaly. The surface is comprised of ROC curves with different buffer lengths. Since VUS calculates the volume under this surface, $l$ doesn't need to be set as a hyperparameter. The main concern of VUS is the label transformation, where false positive points are overestimated than the false negative points.

With the limitation of space, we can not discuss all of the metrics in the TSAD task one by one and we do choose the typical ones. As far as we know, there has not been a perfect metric yet, and it is not certain that there is one. Different metrics have different characteristics. Therefore, suitable metrics should be considered based on specific demands.

## 4 EXPERIMENTS

### 4.1 BASELINE MODELS

We test the most common machine learning and recent SOTA deep learning models in our MTSAD benchmark, which is summarized as follows:

- **Local Outlier Factor (LOF)** (Breunig et al., 2000). LOF measures the local deviation of the density of a given sample with respect to its neighbors.
- **K-Nearest Neighbors (KNN)** (Ramaswamy et al., 2000). KNN views the anomaly score of the input instance as the distance to its $k$-th nearest neighbor.
- **Isolation Forest (IForest)** (Liu et al., 2008). IForest isolates observations by randomly selecting a feature and then randomly selecting a split value between the maximum and minimum values of the selected feature.
- **Long short-term memory (LSTM)** (Malhotra et al., 2015). LSTM is among the family of RNNs (Recurrent Neural Networks) and LSTM (Hochreiter & Schmidhuber, 1997) and can be effectively deployed in the TSAD problem, where the anomalies are detected by the deviation between the predicted and actual ones.
- **LSTM based autoencoder (LSTM-AE)** (Malhotra et al., 2016) reconstructs input sequence and regards samples with high reconstruction errors as anomalies.
- **Deep Support Vector Data Description (DeepSVDD)** (Ruff et al., 2018). DeepSVDD trains a neural network while minimizing the volume of a hypersphere that encloses the network representations of the data, forcing the network to extract the common factors of variation.
- **Deep Autoencoding Gaussian Mixture Model (DAGMM)** (Zong et al., 2018). DAGMM utilizes a deep autoencoder to generate a low-dimensional representation and reconstruction error for each input data point, which is further fed into a Gaussian Mixture Model (GMM).
- **LSTM based variational autoencoder (LSTM-VAE)** (Park et al., 2018) combines the power of both the LSTM-based model and VAE-based model, which learns to encode the input sequence into a lower-dimensional latent space representation and then decodes it back to reconstruct the original sequence. Like LSTM-AE, the reconstruction errors between the input and reconstructed sequences are defined as anomaly scores.
- **Adversarially Generated Model (BeatGAN)** (Zhou et al., 2019). BeatGAN outputs explainable results to pinpoint the anomalous time ticks of an input beat, by comparing them to adversarially generated beats.
- **Copula Based Outlier Detector (COPOD)** (Li et al., 2020). COPOD is a hyperparameter-free, highly interpretable anomaly detection algorithm based on empirical copula models.
- **UnSupervised Anomaly Detection (USAD)** (Audibert et al., 2020). USAD is based on adversely trained autoencoders to isolate anomalies while providing fast training.
- **Anomaly-Transformer** (Xu et al., 2021) Anomaly Transformer is a representation of a series of explicit association modeling works that detect anomalies by association discrepancy between a learned Gaussian kernel and attention weight distribution.
- **Empirical-Cumulative-distribution-based Outlier Detection (ECOD)** (Li et al., 2022). ECOD is a hyperparameter-free, highly interpretable anomaly detection algorithm based on empirical CDF functions. Basically, it uses ECDF to estimate the density of each feature independently, and assumes that the anomaly locates the tails of the distribution.
- **DCdetector** (Yang et al., 2023) DCdetector is a dual attention contrastive representation learning framework whose motivation is similar to anomaly transformer but is concise as it does not contain a specially designed Gaussian kernel or a MinMax learning strategy, nor a reconstruction loss. Contrastive representation learning helps to distinguish anomalies from normal points.

Table 3: MTSAD comparisons on public datasets. Accuracy (Acc), precision (P), recall (R), F1-score (F1), Affiliation precision score (Aff-P), Affiliation recall score (Aff-R), Range-AUC-ROC(R_A_R), Range-AUC-PR(R_A_P), volumes under the surfaces of ROC curve(V_ROC) and volumes under the surfaces of PR curve(V_RR).

| Dataset | Method | Acc | P | R | F1 | Aff-P | Aff-R | R_A_R | R_A_P | V_ROC | V_PR |
|---|---|---|---|---|---|---|---|---|---|---|---|
| MSL | KNN | 93.94 | 47.33 | 90.97 | 62.27 | 70.77 | 9.95 | 55.11 | 37.29 | 55.12 | 37.21 |
| | LOF | 91.86 | 26.42 | 87.69 | 40.61 | 61.83 | 9.78 | 52.15 | 23.60 | 52.08 | 23.40 |
| | IForest | 91.21 | 17.32 | 95.73 | 29.33 | 60.65 | 16.71 | 54.76 | 19.26 | 53.97 | 18.47 |
| | COPOD | 93.27 | 36.69 | 98.51 | 53.46 | 62.43 | 34.28 | 61.89 | 36.08 | 61.73 | 35.83 |
| | ECOD | 93.73 | 40.93 | 98.82 | 57.89 | 67.32 | 33.71 | 63.45 | 39.59 | 63.85 | 39.84 |
| | DeepSVDD | 96.92 | 76.20 | 93.39 | 83.92 | 59.33 | 8.40 | 59.85 | 59.30 | 59.52 | 58.39 |
| | LSTM | 95.86 | 61.35 | 99.17 | 75.81 | 69.34 | 30.46 | 61.56 | 47.96 | 61.38 | 47.73 |
| | LSTM-AE | 89.92 | 4.56 | 100.00 | 8.72 | 48.91 | 100.00 | 85.26 | 42.67 | 85.83 | 43.24 |
| | LSTM-VAE | 89.92 | 4.56 | 100.00 | 8.72 | 48.91 | 100.00 | 85.26 | 42.67 | 85.83 | 43.24 |
| | DAGMM | 92.91 | 95.22 | 34.37 | 50.51 | 60.87 | 42.35 | 58.42 | 18.64 | 57.52 | 18.45 |
| | USAD | 89.89 | 94.86 | 4.28 | 8.18 | 99.54 | 5.56 | 52.42 | 14.22 | 51.86 | 14.28 |
| | BeatGAN | 89.75 | 74.11 | 4.28 | 8.09 | 97.01 | 5.56 | 66.16 | 21.79 | 65.66 | 21.66 |
| | Anomaly-Transformer | 98.69 | 91.92 | 96.03 | 93.93 | 51.76 | 95.98 | 90.04 | 87.87 | 88.2 | 86.26 |
| | DCdetector | 99.06 | 93.69 | 99.69 | 96.6 | 51.84 | 97.39 | 93.17 | 91.64 | 93.15 | 91.66 |
| NIPS_TS_Water | KNN | 96.48 | 99.25 | 94.39 | 96.76 | 89.39 | 2.61 | 68.44 | 87.01 | 66.51 | 86.10 |
| | LOF | 53.65 | 100.00 | 49.56 | 66.28 | 100.00 | 2.52 | 82.24 | 99.08 | 79.35 | 98.81 |
| | IForest | 99.28 | 32.05 | 100.00 | 48.55 | 84.66 | 100.00 | 86.31 | 52.76 | 87.30 | 53.75 |
| | COPOD | 99.28 | 32.05 | 98.32 | 48.35 | 90.84 | 74.86 | 68.98 | 35.55 | 68.81 | 35.42 |
| | ECOD | 99.05 | 10.55 | 97.47 | 19.04 | 84.47 | 99.73 | 61.30 | 17.14 | 63.77 | 19.65 |
| | DeepSVDD | 58.14 | 73.70 | 4.29 | 8.10 | 94.52 | 1.31 | 53.88 | 63.30 | 53.60 | 62.88 |
| | LSTM | 99.29 | 35.47 | 92.53 | 51.28 | 76.58 | 28.04 | 55.43 | 26.27 | 54.82 | 25.10 |
| | LSTM-AE | 99.28 | 32.05 | 97.91 | 48.30 | 85.86 | 66.67 | 81.97 | 48.45 | 78.15 | 44.65 |
| | LSTM-VAE | 99.28 | 32.05 | 97.91 | 48.30 | 85.86 | 66.67 | 81.97 | 48.45 | 78.15 | 44.65 |
| | DAGMM | 98.86 | 36.32 | 10.55 | 16.35 | 75.05 | 10.40 | 71.02 | 5.06 | 71.38 | 5.10 |
| | USAD | 99.29 | 93.80 | 35.21 | 51.20 | 99.45 | 13.64 | 29.55 | 4.23 | 28.74 | 4.31 |
| | BeatGAN | 99.30 | 95.54 | 35.21 | 51.45 | 98.89 | 13.64 | 48.93 | 3.65 | 48.20 | 3.67 |
| | Anomaly-Transformer | 98.26 | 29.96 | 48.63 | 37.08 | 55.65 | 89.12 | 60.74 | 28.17 | 60.48 | 28.02 |
| | DCdetector | 98.23 | 33.46 | 39.05 | 36.04 | 51.67 | 88.96 | 59.12 | 28.84 | 58.50 | 28.25 |
| SMAP | KNN | 93.89 | 52.98 | 98.60 | 68.93 | 58.42 | 10.35 | 51.14 | 35.64 | 51.05 | 35.52 |
| | LOF | 90.94 | 29.77 | 97.98 | 45.67 | 59.95 | 10.12 | 48.91 | 21.59 | 48.74 | 21.37 |
| | IForest | 93.62 | 50.57 | 99.05 | 66.95 | 58.55 | 15.50 | 51.33 | 34.14 | 51.39 | 34.15 |
| | COPOD | 94.02 | 53.80 | 99.03 | 69.72 | 59.52 | 14.42 | 51.51 | 35.94 | 51.57 | 35.94 |
| | ECOD | 94.02 | 53.80 | 99.05 | 69.73 | 59.52 | 14.69 | 51.51 | 35.94 | 51.57 | 35.94 |
| | DeepSVDD | 92.37 | 40.49 | 99.73 | 57.59 | 74.44 | 37.60 | 59.33 | 35.77 | 58.30 | 34.79 |
| | LSTM | 94.00 | 53.73 | 98.88 | 69.62 | 61.55 | 12.01 | 51.35 | 36.11 | 51.36 | 36.06 |
| | LSTM-AE | 94.04 | 54.23 | 98.47 | 69.94 | 63.92 | 13.90 | 51.92 | 36.60 | 51.98 | 36.60 |
| | LSTM-VAE | 93.14 | 47.24 | 98.16 | 63.79 | 65.03 | 21.49 | 52.47 | 33.19 | 52.52 | 33.20 |
| | DAGMM | 93.86 | 98.95 | 52.53 | 68.63 | 58.42 | 58.67 | 45.03 | 12.22 | 45 | 12.25 |
| | USAD | 88.23 | 95.24 | 8.42 | 15.47 | 52.82 | 24.90 | 37.89 | 10.83 | 37.82 | 10.85 |
| | BeatGAN | 94.00 | 98.37 | 53.98 | 69.71 | 74.03 | 62.24 | 44.91 | 12.03 | 44.80 | 12.04 |
| | Anomaly-Transformer | 99.05 | 93.59 | 99.41 | 96.41 | 51.39 | 98.68 | 96.32 | 94.07 | 95.52 | 93.37 |
| | DCdetector | 99.15 | 94.44 | 99.14 | 96.73 | 51.46 | 98.64 | 96.03 | 94.18 | 95.19 | 93.46 |

## 4.2 EXPERIMENT RESULTS ON PUBLIC DATASETS

We evaluate 14 TSAD methods on the 8 public datasets, including 5 classical Machine Learning (ML)-based TSAD methods: KNN, LOF, IForest, COPOD and ECOD; 3 RNN-based TSAD methods: LSTM, LSTM-AE and LSTM-VAE; and 6 Deep Learning-based TSAD methods tailored for time series data: DeepSVDD, DAGMM, USAD, BeatGAN, Anomaly-Transformer and DCdetector, where the state-of-the-art deep models are included. We provide detailed results on three datasets in Table 3, and the table with full results is left in Appendix. Note that we consider all popular and recent proposed metrics for comprehensive evaluation, including accuracy (Acc), precision (P), recall (R), F1-score (F1), Affiliation precision score (Aff-P), Affiliation recall score (Aff-R), Range-AUC-ROC (R_A_R), Range-AUC-PR (R_A_P), volumes under the surfaces of ROC curve (V_ROC), and volumes under the surfaces of PR curve (V_RR).

First, our results show that compared to the classical machine learning-based TSAD methods (especially the simple ML method KNN), the deep learning (DL) counterparts like DeepSVDD do not demonstrate a significant advantage as DeepSVDD shows a similar performance with simple ML on MSL and SMAP datasets with most of the metrics. This conclusion is generally consistent with the findings in previous work (Han et al., 2022), where non-sequential unsupervised TSAD methods are statistically similar to each other. However, we also find that the state-of-the-art deep methods such as Anomaly-Transformer and DCdetector do outperform most of the classical methods as well as some deep methods.

Second, we observe that sequence models can effectively enhance the model's ability to detect anomalies, where LSTM, LSTM-AE, LSTM-VAE and transformer-based models (Anomaly-Transformer and DCdetector) outperform those non-sequential ML-based methods like DeepSVDD, achieving

Table 4: MTSAD comparisons on instance0. Accuracy (Acc), precision (P), recall (R), F1-score (F1), Affiliation precision score (Aff-P), Affiliation recall score (Aff-R), Range-AUC-ROC(R_A_R), Range-AUC-PR(R_A_P), volumes under the surfaces of ROC curve(V_ROC) and volumes under the surfaces of PR curve(V_RR).

| Method | Acc | P | R | F1 | Aff-P | Aff-R | R_A_P | R_A_R | V_PR | V_ROC |
|---|---|---|---|---|---|---|---|---|---|---|
| DAGMM | 98.2 | 61.06 | 24.88 | 35.35 | 75.34 | 60.45 | 87.91 | 15.65 | 86.98 | 15.78 |
| USAD | 78.31 | 7.27 | 84.62 | 13.39 | 76.91 | 85.11 | 15.17 | 84.82 | 15.26 | 84.53 |
| KNN | 69.37 | 6.08 | 100 | 11.46 | 58.06 | 100 | 18.23 | 88.57 | 18.4 | 87.79 |
| LOF | 88.32 | 13.23 | 88.1 | 23.01 | 72.68 | 50 | 17.54 | 89.98 | 17.55 | 89.31 |
| IForest | 94.28 | 24.75 | 92.49 | 39.05 | 67.41 | 89.58 | 8.32 | 85.55 | 8.39 | 84.4 |
| COPOD | 98 | 0 | 0 | 0 | 95.06 | 12.4 | 11.98 | 89.6 | 12.09 | 88.82 |
| ECOD | 98 | 0 | 0 | 0 | 95.88 | 12.4 | 10.13 | 83.68 | 10.32 | 82.4 |
| DeepSVDD | 78.45 | 8.42 | 1 | 15.54 | 59.27 | 1 | 24.61 | 96.32 | 24.46 | 95.82 |
| LSTM | 82 | 8.66 | 84.62 | 15.7 | 83.2 | 84.42 | 15.22 | 85.07 | 15.43 | 84.81 |
| LSTM-AE | 81.33 | 8.37 | 84.62 | 15.23 | 82.59 | 84.51 | 15.45 | 85.02 | 15.68 | 84.8 |
| LSTM-VAE | 81.33 | 8.37 | 84.62 | 15.23 | 82.59 | 84.51 | 15.45 | 85.02 | 15.68 | 84.8 |
| BeatGAN | 79.37 | 7.63 | 84.62 | 14 | 69.82 | 87.32 | 15.2 | 85.02 | 15.42 | 84.76 |
| Anomaly-Transformer | 98.94 | 65.49 | 98.86 | 78.78 | 53.21 | 97.37 | 74.30 | 90.70 | 74.71 | 91.11 |
| DCdetector | 99.01 | 39.95 | 100 | 57.09 | 50.18 | 99.45 | 62.85 | 92.32 | 63.38 | 92.83 |

better results for almost every metric. The most outstanding ones are the Anomaly-Transformer and DCdetector models.

Third, we find that the model based on prediction error (i.e., LSTM) or reconstruction error (i.e., LSTM-AE and LSTM-VAE) has its strengths for sequential TSAD methods. Moreover, LSTM-VAE shows superior to LSTM-AE for all evaluation metrics, indicating that probabilistic modeling of latent space facilitates better capturing of anomalous patterns.

Fourth, the results of different metrics are rather different. We can not conclude that different metrics can help evaluate methods in a consistent way. The main reason is that different designs of metrics indeed lead to different evaluations. How to choose the 'best' metric under a certain situation is an interesting future work.

### 4.3 EXPERIMENTS RESULTS ON AIOPS DATASETS

We also test all 14 anomaly detection methods on our AIOps datasets. Due to the limitation of space, only partial results are shown in Table 4 and Table 7 (in Appendix Section G). We evaluate both classical and deep models for MTSAD. Although most models show extremely good performances on public datasets, things are different in complex real-world datasets. In other words, it seems to be challenging for all the models to handle various real-world data. What is more, the discrepancy among metrics is even larger. We both use default parameters and selected parameters for different instances when evaluating deep models. The performance of the state-of-the-art models is not as good as that on public datasets where we used the well-tuning parameters. So, although deep models are powerful in the MTSAD task, there is still a lot of work to apply them in real scenarios. There are also interesting phenomena in the results which are worth exploring. Some of them are summarized as follows: 1) Different models achieve rather different metric results even on the same instance; 2) Among all the metrics, the variances of R_A_P and R_A_R are the least; 3) The classical methods, such as KNN, seem to perform more robustly among different instances; 4) As real-world data always suffers from missing data, we evaluate different data-filling methods. The results are rather different among different instances. More discussions of the results can be found in Appendix Section G.

### 5 CONCLUSIONS AND ETHICS

In this paper, we design a benchmark specifically for multivariate time series anomaly detection (MTSAD), with a new large-scale real-world AIOps dataset. In order to comprehensively evaluate the performance of different algorithms in MTSAD tasks, we conduct a wide range of experiments in the benchmark with 14 machine learning and deep learning models under 10 metrics, on 8 public datasets as well as our new AIOps datasets. Based on the discussion of the experiments, we unlock insights into the datasets, models, and evaluation metrics.

To promote reproducible results and accelerate the progress of MTSAD research, we have made all the datasets and benchmark implementations publicly available. The datasets and code are both following Apache 2.0 licenses. The data are anonymized by hiding the instances' names and features' names of instances. The data do not contain any personally and individually identifiable information. We have checked for the datasets and made sure the datasets do not have negative impacts on the company, employees, and any other entities.

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

## A    APPENDIX

## B    LIMITATION OF EXISTING DATASETS

### B.0.1    LIMITED COLLECTION OF PUBLIC REAL-WORLD DATASETS

For univariate time series datasets, KPI and Yahoo are the most popular ones with 58 and 367 curves, respectively. Although TSB-UAD (Paparrizos et al., 2022b) provides 13766 time series with labeled anomalies, 10828 of them are synthetic data where different strategies are used to increase detecting difficulty. However, one main concern is that, with data transformed, the labels usually remain unchanged which may result in wrong labels. What's worse, if data is created artificially, it is hard to say the data is natural and the performance of models on it may be manipulated through information given by generation rules/processing. That is also why real-world data matters in the TSAD tasks.

For multivariate time series datasets, although more public datasets are available, the number of samples is less as one point is contained with several dimensions. The size of multivariate time series datasets is even less. Performances of models vary among different datasets. It is urgent to collect more real data for MTSAD.

### B.0.2    CONFUSING GROUND TRUTH LABELS

Another flaw of existing datasets is that mislabels happen in all these datasets. It may be mainly due to the difficulty for experts to check for every label. However, non-negligible mislabels do hurt the evaluation of models and even lead to wrong research directions.

### B.0.3    LIMITED TYPES OF TIME SERIES ANOMALIES

Time series anomalies can be roughly classified as point-wise anomalies and pattern-wise anomalies (Lai et al., 2021) where point-wise anomalies contain global and contextual outliers, pattern-wise anomalies contain shapelet outliers, seasonal outliers and trend outliers. However, most of the anomalies in public datasets are peaks or valleys. This is also part of the reason why random guesses can achieve an even better score than most of the well-designed anomaly detection models (Doshi et al., 2022; Kim et al., 2022). Lack of a good performance metric also causes such results which we will discuss in Section 3.3. Seasonal outliers and trend anomalies are rare which is also why many synthetic datasets, including the Yahoo dataset, datasets in TSB-UAD and UCR dataset, are generated for univariate TSAD.

Actually, the above types of outliers are also mainly set for single/univariate time series. When more than one time series is considered, it will be much more complicated as a change of relationships among different time series or channels may also lead to anomalies. Different from univariate time series, it is usually too difficult to generate synthetic MTSAD datasets as the types of anomalies in multivariate time series are hard to define and then there are few rules that can be followed for a generation. Thus, real-world large-scale multivariate time series datasets with reliable labels are in urgent need.

## C    CONFUSING LABELS IN EXISTING DATASET

**Confusing Labels in KPI**    Figure 4(a) mainly shows peak-type anomalies in time series. However, it is confusing that the peak around index 97900 is labeled as abnormal, while a higher peak in around index 14850 is labeled as normal. A similar thing happens in valley-type anomalies. As shown in Figure 4(b), while valleys in indexes around 7300 and 15680 are both labeled as anomalies, a deeper valley in index 10020 is considered as normal.

**Confusing Labels in Yahoo**    Figure 5 gives some examples where the labels of ground truth are confusing. The beginning points of A1_28, A1_38, and A1_55 are abnormal but the labels are not abnormal as shown in Figure 5(b), 5(c) respectively. In A1_38 (Figure 5(b)), the points around index 646 are too high to be normal but they are labeled as normal. In A1_55 (Figure 5(c)), only one point (whose index is 1206) is labeled as abnormal when a new pattern happens at around 1200. However, in A1_32 (Figure 5(a)), if the change in index around 1221 is a new pattern, further more than one point is labeled as abnormal which is different from A1_55. If all the points in this 'new pattern' are considered as abnormal, all of them should be labeled as anomalies. Such inconsistency makes the labels confusing.

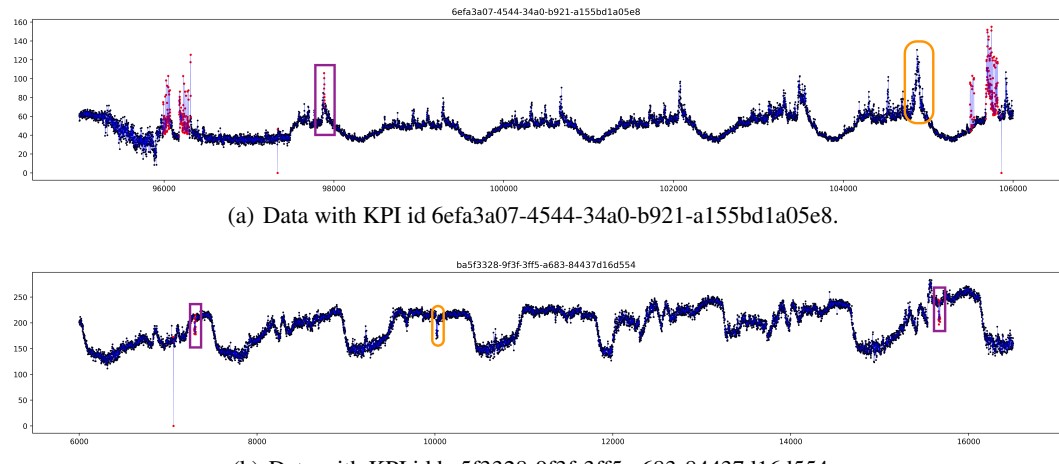

(a) Data with KPI id 6efa3a07-4544-34a0-b921-a155bd1a05e8.

(b) Data with KPI id ba5f3328-9f3f-3ff5-a683-84437d16d554.

Figure 4: Confusing labels in KPI dataset. Normal points are black and abnormal points are red. The lines show changes among points are blue.

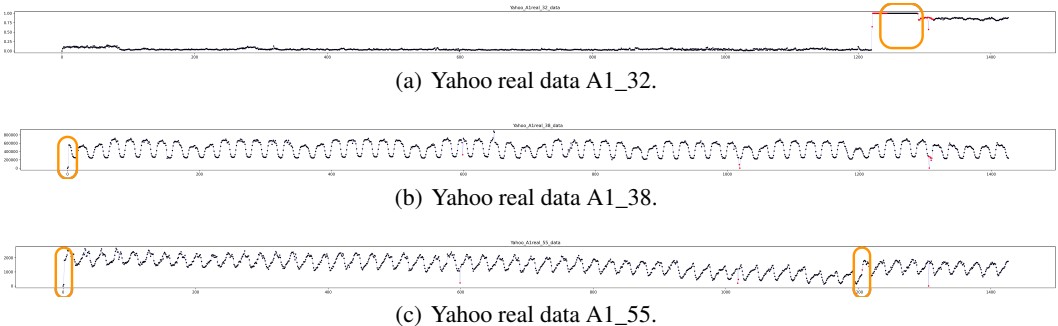

(a) Yahoo real data A1_32.

(b) Yahoo real data A1_38.

(c) Yahoo real data A1_55.

Figure 5: Confusing labels in Yahoo dataset. Normal points are black and abnormal points are red. The lines show changes among points are blue.

**Confusing Labels in SMAP**   Figure 6 shows an example of anomalies in the NASA-SMAP test dataset from index 4600 to 4800. It is hard to understand why the points in the window from 4690 to 4770 are all abnormal. On the left of the window, no peaks appear which is just the same as the window from 4620 to 4650. However, the labels are not the same.

Other MTSAD datasets have similar flaws with labels. We will not show all of them here.

## D  ILLUSTRATIONS OF REAL-WORLD AIOPS DATASETS.

Figure 7 shows the architecture of the real-time data warehouse where the real-world datasets are collected in this paper. Figure 8 shows a real-world case of multivariate time series anomaly detection from a data warehouse instance.

## E  DETAILED EVALUATION METRICS

**F1-Score with Point Adjustment**   This metric is proposed by  (Xu et al., 2018; Audibert et al., 2020). It works as follows: if one anomaly point is correctly detected in the ground truth anomaly segment, all the points in such segment will be considered as correctly detected. Then F1-score is calculated with such adjusted predictions. F1-PA is designed with the alert that one detected anomaly shows errors in the system sufficiently. However, such a metric has a high possibility of overestimating the performance of models and does not consider the information of anomaly events. Actually, with the F1-PA metric, even random guess gains SOTA performance (Kim et al., 2022; Doshi et al., 2022).

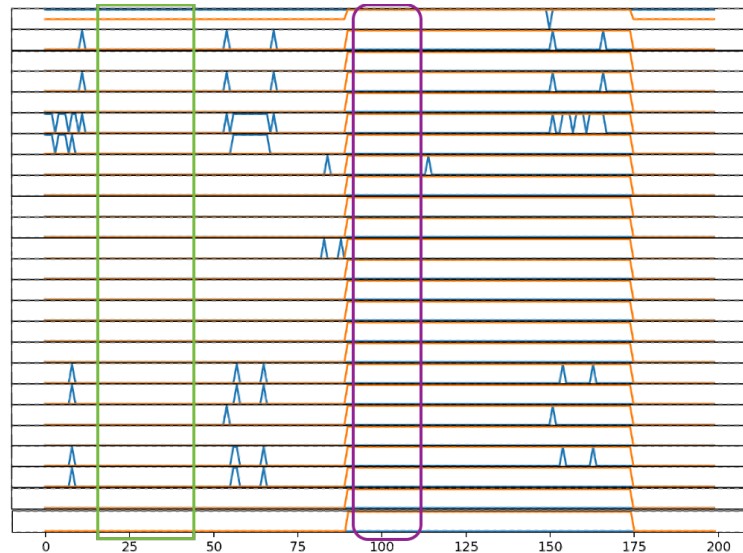

Figure 6: Confusing labels in SMAP dataset. In each dimension, the blue line is the original values and the orange line is the labels.

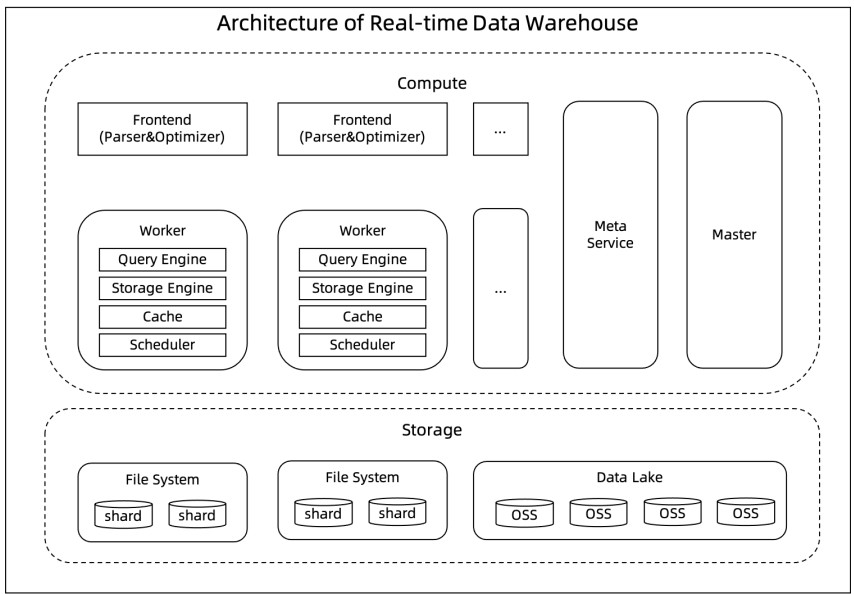

Figure 7: Architecture of the real-time data warehouse.

Besides the original F1-score with point adjustment, there are also several variants of F1-PA. For example, F1-PA%K (Kim et al., 2022) applies point adjustment only when the ratio of the number of correctly detected points is larger than the %K of anomaly length. K is a threshold. It indeed helps relieve over-optimistic in F1-PA but does not solve the problem of F1-PA. The gut issue of F1-PA is that it considers anomalies from a point-wise view. However, in the time series field, it is hard to say reasonable to consider a single point as a sample. A more natural way is to define something like anomaly events to gain information.

**Composite F1-score** It is a metric taking event-wise anomaly into account (Garg et al., 2021) but still keeps the main design of point adjustment. Specially, it takes a point-wise precision with point adjustment and an event-wise recall. The formalization is shown as follows.

$$Pr_t = \frac{TP_t}{TP_t + FP_t} \quad \text{and} \quad Rec_e = \frac{TP_e}{TP_e + FN_e}, \tag{2}$$

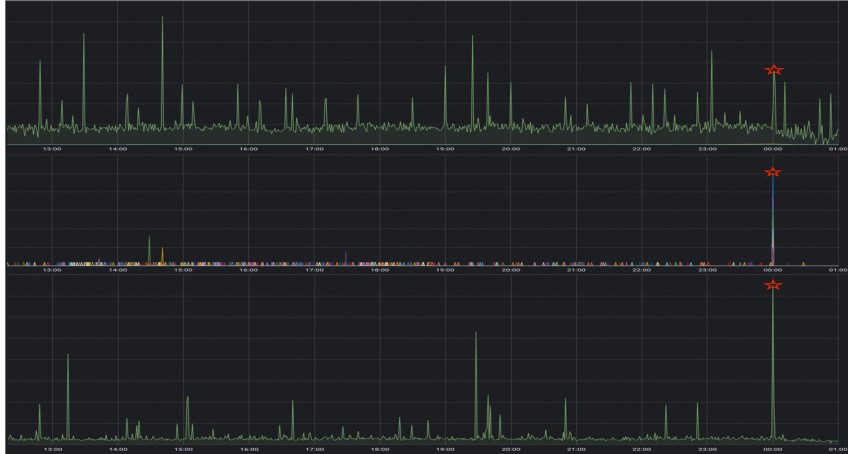

Figure 8: Real-world case of multivariate time series anomaly detection from a data warehouse instance.

where $\text{TP}_t$ and $\text{FP}_t$ are the numbers of TP and FP points respectively, $\text{TP}_e$ and $\text{FN}_e$ are the number of TP and FN events respectively. $\text{TP}_e$ is the number of true events for which at least one point is detected rightly. The other true events are counted under $\text{FN}_e$. This metric doesn't differentiate the locations of false positive events and over punish missing detection of single point events. What's more, it is not very persuasive that precision and recall should be defined in different views. Actually, there are metrics taking the position of results into consideration, like NAB score (Lavin & Ahmad, 2015), SPD score (Doshi et al., 2022). Recently, affiliation metric is proposed with pure event-view to deal with the above challenges.

**Affiliation Score**  Affiliation (Huet et al., 2022) is a metric with an intuitive interpretation where both precision and recall are calculated based on the distance between ground truth and prediction events. Event distance is defined through point sets by Hausdorff distance (Dubuisson & Jain, 1994) and precision/recall is set by individual probability based on event distance normalized by affiliation zone. Affiliation is proven to be robust against adversary strategies. It is novel to measure event distance by Hausdorff distance and exquisite to draw individual probability into precision and recall. However, the affiliation zone has a huge influence on the final score. With little improvement in precision, the bigger size of the zone results in a higher score in a non-negligible degree. What's more, all the prediction events in the zone contribute to the final score even when they are false positives. Actually, if a prediction event is far from the ground truth, it should be punished. Another phenomenon caused by zone splitation is with a high tolerance for false positives but a tolerance low for false negative points.

**Volume Under the Surface (VUS) Metric**  Besides the above metrics based on the F1-score, there are also metrics based on the receiver operator characteristic (ROC) curve and the area under the curve (AUC). The original ROC and AUC are based on point-wise detection. However, as discussed above, such point-wise type metrics introduce unavoidable shortcomings in range-based anomalies by mapping discrete labels into continuous data. That is why the event-based F1-score appears. VUS metric extends the AUC-based measures to account for range-based anomalies. The key designs are the label transformation technique and volume under the surface metric. For label transformation, with a buffer length, the binary label is extended into a continuous value. Given buffer length $l$, the positions $s, e \in [0, |label|]$ the beginning and end indexes of a labeled (range) anomaly, the formalization of the continuous $label_r$ is set as follows:

$$\forall i \in [0, |label|], \quad label_{li} = \begin{cases} (1 - \frac{|s-i|}{l})^{\frac{1}{2}} & \text{if: } s - \frac{l}{2} \leq i < s \\ 1 & \text{if: } s \leq i < e \\ (1 - \frac{|e-i|}{l})^{\frac{1}{2}} & \text{if: } e \leq i < e + \frac{l}{2} \\ 0 & \text{if: } i < s \text{ or } e < i \end{cases}.$$

The surface is comprised of ROC curves with different buffer lengths. Thus, $l$ doesn't need to be set as a hyperparameter. The main concern of VUS is with label transformation, the false positive points are overestimated than the false negative points. It is not sure if it is better for specific situations.

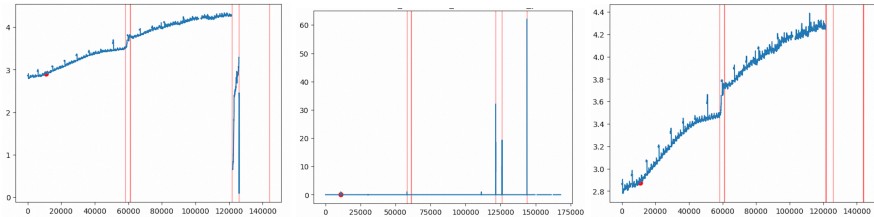

Figure 9: Visualization of part of metrics of Instance 18 where the red line instructs anomalies happening and the vertical axis is normalized.

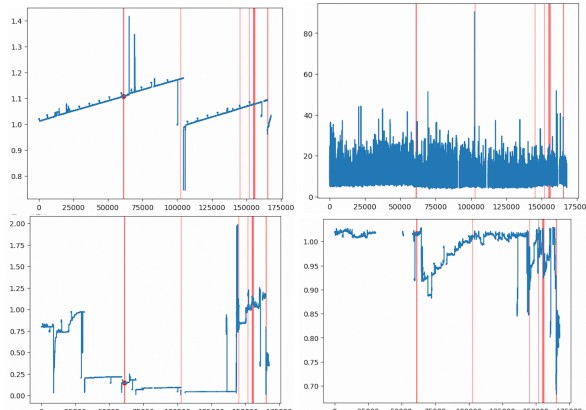

Figure 10: Visualization of part of metrics of Instance 23 where the red line instructs anomalies happening and the vertical axis is normalized.

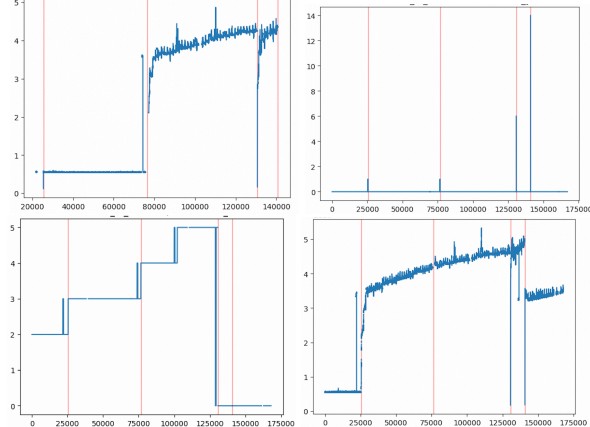

Figure 11: Visualization of part of metrics of Instance 28 where the red line instructs anomalies happening and the vertical axis is normalized.

Here, we show the full results of different methods on 8 datasets with various metrics in Table 5 and Table 6.

# F  VISUALIZATION OF REAL-WORLD AIOPS DATASET

We have proposed real-world multivariate time series datasets from the AIOps system of the real-time data warehouse. In this section, we would like to show some visualization of the instances to make it more intuitive to the users. Figure 9, Figure 10, and Figure 11 show some of the metrics of Instance 18, Instance 23, and Instance 28, respectively, where the red line represents anomalies. We visualize these different instances to demonstrate the complexity of anomalies in multivariate time series.

# G   Detailed Experiment Results and Discussion

In this section, we summarize more experiment results of those instances in the AIOps datasets. To compare the performance of different methods, we evaluate different methods with hyperparameter selection and summarize the results in Table 7. For the processing of missing data, Table 8 shows the experiment results on part of the instances with filling mean for missing data, where the abbreviations of the evaluation metrics are accuracy, precision, recall, F1-score, affiliation precision, affiliation recall (Huet et al., 2022), Range_AUC_ROC, Range_AUC_PR, VUS_ROC, VUS_PR, AUC_PR, and AUC_ROC (Paparrizos et al., 2022a) in order. Besides, we also evaluate other methods for filling missing data with zero interpolation and linear interpolation as shown in Table 9 and Table 10, respectively.

From the results, we have discovered some interesting phenomena.

- Different models achieve rather different metric results even in the same instance. What's more, the order of the performance on different metrics is also inconsistent. For example, although the state-of-the-art deep models Anomaly-Transformer and DCdetector do not gain a good performance on F1 with point adjustment, they achieve the best ones in V_ROC and V_PR. This may be mainly because the V_ROC and V_PR metrics are more sensitive for detection in the recall direction and take the recall and the precision balanced. The ECOD model performs almost the best in Precision with point adjustment (P metric) and almost worst with Recall with point adjustment (R metric). At the same time, it also gains almost the worst score in V_PR. Such inconsistency among metrics also indicates the importance of the choice of the proper metric for a certain situation.
- Among all the metrics, the variances of R_A_P and R_A_R are the least. For instance, in the results of instance 38, the variances of different models on metric R and metric R are extremely large. However, we do not see a huge gap among models on the A_PR and A_R metrics. We are not sure which is expected in a general case. Is it reasonable to have such a large gap among the models? Actually, the detection differences among the models may just be a few points. However, is A_PR or A_R a good one? If the detection purpose is to choose the best method, such a "robust" metric may not be a good choice. We will leave this as an interesting future work.
- The classical methods, such as KNN, seem to perform more robustly among different instances. For example, with filling the mean for missing data, the P metric does not show a large variance among instances (most ranging from 0.09 2.19). However, the deep method, such as Anomaly-Transformer, can range from 0 to 60.28. Part of the reason is that we do not make parameter adjustments for each instance which has an influence on deep models, while classical methods are more robust with hyperparameters.
- As real-world data always suffers from missing data, we evaluate different data-filling methods. The results are rather different among different instances. We take instance14 and instance44 as examples. To clarify the discussion, we first consider the "robust" A_R metric. The results of instance14 are similar with different filling methods. That is, different models show similar performances with different filling methods. However, the thing is very different for instance 44. With the filling mean method, USAD and KNN achieve 89.04 and 92.66, respectively. While, with the filling linear interpolation method, they achieve only 32.54 and 40.40, respectively. What's more, BeatGAN gains 89.01 score with filling mean and only 34.48 with filling linear interpolation.
- The models show rather different performances on Recall and Precision. For example, with filling zero, KNN and LOF both gain 100 (100 percent) recall for instance 14 but with only 1.75 for precision. It is also common in reality that in different situations, we take different views into consideration. Sometimes, recall is important as a missing anomaly may lead to a huge loss. In other situations, precision is more important as too many anomaly alarms are not acceptable. However, how to choose or design a metric to apply in real situations is extremely important and challenging.

There are still many works for time series anomaly detection in the real world. And the gap between public datasets/metrics and real-world application evaluations is still large. We hope our work can inspire more interest in exploring real-world applications.

Table 5: MTSAD comparisons on all public datasets - part 1.

| Dataset | Method | Acc | P | R | F1 | Aff-P | Aff-R | R_A_R | R_A_P | V_ROC | V_PR |
|---|---|---|---|---|---|---|---|---|---|---|---|
| MSL | KNN | 93.94 | 47.33 | 90.97 | 62.27 | 70.77 | 9.95 | 55.11 | 37.29 | 55.12 | 37.21 |
| | LOF | 91.86 | 26.42 | 87.69 | 40.61 | 61.83 | 9.78 | 52.15 | 23.60 | 52.08 | 23.40 |
| | IForest | 91.21 | 17.32 | 95.73 | 29.33 | 60.65 | 16.71 | 54.76 | 19.26 | 53.97 | 18.47 |
| | COPOD | 93.27 | 36.69 | 98.51 | 53.46 | 62.43 | 34.28 | 61.89 | 36.08 | 61.73 | 35.83 |
| | ECOD | 93.73 | 40.93 | 98.82 | 57.89 | 67.32 | 33.71 | 63.45 | 39.59 | 63.85 | 39.84 |
| | DeepSVDD | 96.92 | 76.20 | 93.39 | 83.92 | 59.33 | 8.40 | 59.85 | 59.30 | 59.52 | 58.39 |
| | LSTM | 95.86 | 61.35 | 99.17 | 75.81 | 69.34 | 30.46 | 61.56 | 47.96 | 61.38 | 47.73 |
| | LSTM-AE | 89.92 | 4.56 | 100.00 | 8.72 | 48.91 | 100.00 | 85.26 | 42.67 | 85.83 | 43.24 |
| | LSTM-VAE | 89.92 | 4.56 | 100.00 | 8.72 | 48.91 | 100.00 | 85.26 | 42.67 | 85.83 | 43.24 |
| | DAGMM | 92.91 | 95.22 | 34.37 | 50.51 | 60.87 | 42.35 | 58.42 | 18.64 | 57.52 | 18.45 |
| | USAD | 89.89 | 94.86 | 4.28 | 8.18 | 99.54 | 5.56 | 52.42 | 14.22 | 51.86 | 14.28 |
| | BeatGAN | 89.75 | 74.11 | 4.28 | 8.09 | 97.01 | 5.56 | 66.16 | 21.79 | 65.66 | 21.66 |
| | Anomaly-Transformer | 98.69 | 91.92 | 96.03 | 93.93 | 51.76 | 95.98 | 90.04 | 87.87 | 88.2 | 86.26 |
| | DCdetector | 99.06 | 93.69 | 99.69 | 96.6 | 51.84 | 97.39 | 93.17 | 91.64 | 93.15 | 91.66 |
| NIPS_TS_Ccard | KNN | 99.67 | 39.01 | 21.01 | 27.32 | 74.96 | 28.40 | 55.57 | 37.74 | 55.51 | 37.20 |
| | LOF | 99.64 | 0.00 | 0.00 | 0.00 | 74.84 | 18.17 | 54.08 | 12.16 | 54.01 | 12.48 |
| | IForest | 99.81 | 13.45 | 26.55 | 17.86 | 60.38 | 52.84 | 51.44 | 15.39 | 51.51 | 14.78 |
| | COPOD | 99.82 | 17.49 | 35.14 | 23.35 | 62.52 | 57.23 | 51.32 | 14.80 | 51.46 | 14.49 |
| | ECOD | 99.83 | 17.04 | 39.58 | 23.82 | 64.79 | 62.22 | 51.13 | 13.61 | 51.33 | 13.48 |
| | DeepSVDD | 99.74 | 0.45 | 0.65 | 0.53 | 54.28 | 32.41 | 52.22 | 8.29 | 52.20 | 8.38 |
| | LSTM | 99.85 | 22.97 | 55.43 | 32.48 | 69.08 | 73.53 | 51.05 | 14.26 | 51.53 | 14.95 |
| | LSTM-AE | 99.72 | 7.42 | 8.54 | 7.94 | 56.39 | 67.03 | 50.87 | 12.80 | 50.91 | 12.25 |
| | LSTM-VAE | 99.79 | 21.34 | 33.54 | 26.09 | 59.77 | 84.66 | 51.66 | 18.08 | 52.45 | 18.71 |
| | DAGMM | 99.73 | 0.59 | 0.45 | 0.51 | 52.39 | 23.8 | 76.5 | 10.02 | 76.16 | 9.71 |
| | Anomaly-Transformer | 99.66 | 0 | 0 | 0 | 50.76 | 37.14 | 52.51 | 11.91 | 52.46 | 11.65 |
| | DCdetector | 99.73 | 0.65 | 0.45 | 0.53 | 46.51 | 23.30 | 52.52 | 9.93 | 52.46 | 9.08 |
| | USAD | 99.78 | 22.50 | 16.14 | 18.80 | 62.13 | 9.71 | 86.97 | 23.26 | 86.73 | 22.08 |
| | BeatGAN | 99.85 | 53.54 | 23.77 | 32.92 | 74.02 | 24.17 | 81.83 | 14.42 | 82.31 | 13.90 |
| NIPS_TS_Swan | KNN | 88.16 | 64.87 | 98.38 | 78.18 | 85.80 | 88.02 | 78.40 | 75.07 | 79.04 | 75.40 |
| | LOF | 67.38 | 0.00 | 0.00 | 0.00 | 44.84 | 98.67 | 47.40 | 14.20 | 47.26 | 14.06 |
| | IForest | 86.55 | 58.76 | 99.95 | 74.01 | 66.09 | 93.02 | 89.98 | 78.54 | 88.28 | 77.17 |
| | COPOD | 86.47 | 58.50 | 100.00 | 73.82 | 76.24 | 100.00 | 91.63 | 79.27 | 91.62 | 79.26 |
| | ECOD | 86.44 | 58.50 | 99.83 | 73.77 | 48.12 | 80.20 | 69.73 | 61.93 | 72.23 | 63.92 |
| | DeepSVDD | 86.49 | 58.60 | 99.97 | 73.89 | 53.87 | 97.44 | 90.99 | 78.97 | 90.52 | 78.59 |
| | LSTM | 87.78 | 63.22 | 98.90 | 77.14 | 82.21 | 89.60 | 80.05 | 74.48 | 81.13 | 75.10 |
| | LSTM-AE | 86.48 | 58.58 | 99.98 | 73.88 | 56.05 | 98.37 | 78.97 | 69.22 | 78.95 | 69.20 |
| | LSTM-VAE | 86.47 | 58.50 | 100.00 | 73.82 | 76.24 | 100.00 | 91.63 | 79.27 | 91.62 | 79.26 |
| | DAGMM | 86.37 | 99.09 | 58.71 | 73.74 | 54.64 | 1.06 | 91.88 | 91.05 | 91.32 | 90.01 |
| | USAD | 86.45 | 99.39 | 58.78 | 73.87 | 68.00 | 0.66 | 93.64 | 93.46 | 91.24 | 91.41 |
| NIPS_TS_Syn_Mulvar | KNN | 79.92 | 8.55 | 100.00 | 15.75 | 55.95 | 100.00 | 67.73 | 40.33 | 69.91 | 42.20 |
| | LOF | 79.43 | 6.32 | 99.82 | 11.89 | 53.56 | 99.29 | 65.86 | 36.17 | 67.94 | 37.97 |
| | IForest | 79.55 | 6.88 | 100.00 | 12.87 | 64.89 | 100.00 | 64.28 | 35.35 | 65.74 | 36.25 |
| | COPOD | 78.42 | 1.91 | 90.32 | 3.75 | 53.24 | 95.51 | 62.80 | 29.26 | 63.00 | 29.29 |
| | ECOD | 78.56 | 2.53 | 94.28 | 4.93 | 51.53 | 98.29 | 63.76 | 30.65 | 64.52 | 31.15 |
| | DeepSVDD | 79.08 | 4.74 | 100.00 | 9.05 | 53.23 | 100.00 | 67.32 | 34.26 | 69.33 | 36.22 |
| | LSTM | 79.13 | 4.94 | 100.00 | 9.42 | 52.70 | 100.00 | 68.66 | 35.73 | 70.32 | 37.26 |
| | LSTM-AE | 78.43 | 2.14 | 87.85 | 4.18 | 50.26 | 99.07 | 65.50 | 32.16 | 65.93 | 32.41 |
| | LSTM-VAE | 78.25 | 1.40 | 79.10 | 2.75 | 50.08 | 99.12 | 65.21 | 31.98 | 65.26 | 31.89 |
| | DAGMM | 78.31 | 90.94 | 1.37 | 2.70 | 74.05 | 0.59 | 99.99 | 99.98 | 97.33 | 95.7 |
| | USAD | 78.04 | 48.8 | 7.88 | 13.61 | 50.49 | 8.31 | 99.98 | 99.98 | 96.53 | 95.23 |
| NIPS_TS_Water | KNN | 96.48 | 99.25 | 94.39 | 96.76 | 89.39 | 2.61 | 68.44 | 87.01 | 66.51 | 86.10 |
| | LOF | 53.65 | 100.00 | 49.56 | 66.28 | 100.00 | 2.52 | 82.24 | 99.08 | 79.35 | 98.81 |
| | IForest | 99.28 | 32.05 | 100.00 | 48.55 | 84.66 | 100.00 | 86.31 | 52.76 | 87.30 | 53.75 |
| | COPOD | 99.28 | 32.05 | 98.32 | 48.35 | 90.84 | 74.86 | 68.98 | 35.55 | 68.81 | 35.42 |
| | ECOD | 99.05 | 10.55 | 97.47 | 19.04 | 84.47 | 99.73 | 61.30 | 17.14 | 63.77 | 19.65 |
| | DeepSVDD | 58.14 | 73.70 | 4.29 | 8.10 | 94.52 | 1.31 | 53.88 | 63.30 | 53.60 | 62.88 |
| | LSTM | 99.29 | 35.47 | 92.53 | 51.28 | 76.58 | 28.04 | 55.43 | 26.27 | 54.82 | 25.10 |
| | LSTM-AE | 99.28 | 32.05 | 97.91 | 48.30 | 85.86 | 66.67 | 81.97 | 48.45 | 78.15 | 44.65 |
| | LSTM-VAE | 99.28 | 32.05 | 97.91 | 48.30 | 85.86 | 66.67 | 81.97 | 48.45 | 78.15 | 44.65 |
| | DAGMM | 98.86 | 36.32 | 10.55 | 16.35 | 75.05 | 10.40 | 71.02 | 5.06 | 71.38 | 5.10 |
| | USAD | 99.29 | 93.80 | 35.21 | 51.20 | 99.45 | 13.64 | 29.55 | 4.23 | 28.74 | 4.31 |
| | BeatGAN | 99.30 | 95.54 | 35.21 | 51.45 | 98.89 | 13.64 | 48.93 | 3.65 | 48.20 | 3.67 |
| | Anomaly-Transformer | 98.26 | 29.96 | 48.63 | 37.08 | 55.65 | 89.12 | 60.74 | 28.17 | 60.48 | 28.02 |
| | DCdetector | 98.23 | 33.46 | 39.05 | 36.04 | 51.67 | 88.96 | 59.12 | 28.84 | 58.50 | 28.25 |
| PSM | KNN | 94.84 | 95.31 | 91.98 | 93.62 | 93.33 | 6.38 | 73.90 | 84.71 | 70.89 | 82.86 |
| | LOF | 85.64 | 99.96 | 77.41 | 87.25 | 89.22 | 1.69 | 76.08 | 93.79 | 74.49 | 92.97 |
| | IForest | 78.36 | 22.04 | 100.00 | 36.12 | 55.22 | 100.00 | 87.32 | 59.95 | 87.33 | 59.97 |
| | COPOD | 78.36 | 22.04 | 100.00 | 36.12 | 55.22 | 100.00 | 87.32 | 59.95 | 87.33 | 59.97 |
| | ECOD | 78.36 | 22.04 | 100.00 | 36.12 | 55.22 | 100.00 | 87.32 | 59.95 | 87.33 | 59.97 |
| | DeepSVDD | 93.14 | 92.93 | 88.12 | 90.46 | 86.52 | 7.43 | 73.99 | 82.53 | 71.55 | 80.93 |
| | LSTM | 95.25 | 82.93 | 99.96 | 90.65 | 79.89 | 89.90 | 90.54 | 86.74 | 89.98 | 86.30 |
| | LSTM-AE | 92.84 | 79.39 | 99.72 | 88.40 | 77.36 | 43.84 | 81.79 | 80.16 | 83.03 | 81.03 |
| | LSTM-VAE | 97.10 | 92.83 | 99.80 | 96.19 | 85.35 | 65.86 | 96.82 | 95.86 | 95.73 | 95.17 |
| | Anomaly-Transformer | 98.68 | 96.94 | 97.81 | 97.37 | 55.35 | 80.28 | 91.83 | 93.03 | 88.71 | 90.71 |
| | DCdetector | 98.95 | 97.14 | 98.74 | 97.94 | 54.71 | 82.93 | 91.55 | 92.93 | 88.41 | 90.58 |

Table 6: MTSAD comparisons on all public datasets - part 2.

| Dataset | Method | Acc | P | R | F1 | Aff-P | Aff-R | R_A_R | R_A_P | V_ROC | V_PR |
|---|---|---|---|---|---|---|---|---|---|---|---|
| SMAP | KNN | 93.89 | 52.98 | 98.60 | 68.93 | 58.42 | 10.35 | 51.14 | 35.64 | 51.05 | 35.52 |
| | LOF | 90.94 | 29.77 | 97.98 | 45.67 | 59.95 | 10.12 | 48.91 | 21.59 | 48.74 | 21.37 |
| | IForest | 93.62 | 50.57 | 99.05 | 66.95 | 58.55 | 15.50 | 51.33 | 34.14 | 51.39 | 34.15 |
| | COPOD | 94.02 | 53.80 | 99.03 | 69.72 | 59.52 | 14.42 | 51.51 | 35.94 | 51.57 | 35.94 |
| | ECOD | 94.02 | 53.80 | 99.05 | 69.73 | 59.52 | 14.69 | 51.51 | 35.94 | 51.57 | 35.94 |
| | DeepSVDD | 92.37 | 40.49 | 99.73 | 57.59 | 74.44 | 37.60 | 59.33 | 35.77 | 58.30 | 34.79 |
| | LSTM | 94.00 | 53.73 | 98.88 | 69.62 | 61.55 | 12.01 | 51.35 | 36.11 | 51.36 | 36.06 |
| | LSTM-AE | 94.04 | 54.23 | 98.47 | 69.94 | 63.92 | 13.90 | 51.92 | 36.60 | 51.98 | 36.60 |
| | LSTM-VAE | 93.14 | 47.24 | 98.16 | 63.79 | 65.03 | 21.49 | 52.47 | 33.19 | 52.52 | 33.20 |
| | DAGMM | 93.86 | 98.95 | 52.53 | 68.63 | 58.42 | 58.67 | 45.03 | 12.22 | 45 | 12.25 |
| | USAD | 88.23 | 95.24 | 8.42 | 15.47 | 52.82 | 24.90 | 37.89 | 10.83 | 37.82 | 10.85 |
| | BeatGAN | 94.00 | 98.37 | 53.98 | 69.71 | 74.03 | 62.24 | 44.91 | 12.03 | 44.80 | 12.04 |
| | Anomaly-Transformer | 99.05 | 93.59 | 99.41 | 96.41 | 51.39 | 98.68 | 96.32 | 94.07 | 95.52 | 93.37 |
| | DCdetector | 99.15 | 94.44 | 99.14 | 96.73 | 51.46 | 98.64 | 96.03 | 94.18 | 95.19 | 93.46 |
| SMD | KNN | 91.95 | 90.88 | 41.40 | 56.89 | 92.23 | 3.83 | 58.59 | 62.47 | 57.98 | 61.76 |
| | LOF | 79.36 | 96.94 | 27.57 | 42.93 | 88.19 | 1.68 | 60.69 | 74.76 | 60.04 | 73.95 |
| | IForest | 97.49 | 42.35 | 93.97 | 58.39 | 64.30 | 13.65 | 59.92 | 33.61 | 58.99 | 32.65 |
| | COPOD | 96.78 | 24.70 | 91.95 | 38.94 | 61.03 | 26.13 | 68.55 | 32.91 | 67.67 | 32.05 |
| | ECOD | 96.81 | 24.29 | 95.50 | 38.73 | 62.58 | 25.26 | 72.43 | 36.52 | 72.17 | 36.26 |
| | DeepSVDD | 97.36 | 50.57 | 78.37 | 61.47 | 72.99 | 10.66 | 61.33 | 39.38 | 60.88 | 38.92 |
| | LSTM | 98.84 | 76.10 | 94.99 | 84.50 | 83.84 | 15.66 | 59.06 | 50.90 | 58.74 | 50.48 |
| | LSTM-AE | 97.16 | 65.83 | 68.27 | 67.03 | 80.45 | 15.63 | 64.10 | 49.97 | 63.68 | 49.56 |
| | LSTM-VAE | 96.96 | 82.35 | 63.50 | 71.71 | 87.07 | 16.18 | 63.98 | 58.57 | 63.03 | 57.65 |
| | DAGMM | 96.86 | 88.78 | 28.05 | 42.63 | 69.55 | 16.4 | 63.69 | 9.67 | 63.06 | 9.62 |
| | Anomaly-Transformer | 99.16 | 88.47 | 92.28 | 90.33 | 58.94 | 91.79 | 76.57 | 72.76 | 76.67 | 72.88 |
| | DCdetector | 98.86 | 83.59 | 91.1 | 87.18 | 52.72 | 93.8 | 78.04 | 71.96 | 75.15 | 69.23 |
| | USAD | 96.45 | 89.42 | 16.51 | 27.87 | 85.03 | 3.81 | 57.98 | 10.12 | 57.34 | 10.09 |
| | BeatGAN | 97.44 | 80.77 | 50.36 | 62.04 | 90.00 | 28.30 | 76.83 | 14.59 | 76.28 | 14.47 |
| Ave. | KNN | 95.19 | 65.89 | 69.27 | 62.43 | 77.15 | 11.03 | 57.77 | 52.03 | 57.23 | 51.56 |
| | LOF | 83.09 | 50.63 | 52.56 | 39.10 | 71.66 | 8.45 | 59.61 | 46.24 | 58.84 | 46.00 |
| | IForest | 96.28 | 31.15 | 83.06 | 44.22 | 65.71 | 39.74 | 60.75 | 31.03 | 60.63 | 30.76 |
| | COPOD | 96.63 | 32.95 | 84.59 | 46.76 | 67.27 | 41.38 | 60.45 | 31.06 | 60.25 | 30.75 |
| | ECOD | 96.69 | 29.32 | 86.08 | 41.84 | 67.74 | 47.12 | 59.96 | 28.56 | 60.54 | 29.03 |
| | DeepSVDD | 88.91 | 48.28 | 55.29 | 42.32 | 71.11 | 18.08 | 57.32 | 41.21 | 56.90 | 40.67 |
| | LSTM | 97.57 | 49.92 | 88.20 | 62.74 | 72.08 | 31.94 | 55.69 | 35.10 | 55.57 | 34.86 |
| | LSTM-AE | 96.02 | 32.82 | 74.64 | 40.39 | 67.11 | 52.65 | 66.82 | 38.10 | 66.11 | 37.26 |
| | LSTM-VAE | 95.82 | 37.51 | 78.62 | 43.72 | 69.33 | 57.80 | 67.07 | 40.19 | 66.40 | 39.49 |
| | DAGMM | 94.89 | 53.31 | 20.99 | 29.77 | 59.03 | 25.35 | 65.60 | 18.68 | 65.26 | 18.44 |
| | USAD | 94.72 | 79.16 | 16.11 | 24.30 | 79.79 | 11.52 | 52.96 | 12.53 | 52.49 | 12.32 |
| | BeatGAN | 96.06 | 80.46 | 33.52 | 44.84 | 86.79 | 26.78 | 63.73 | 13.29 | 63.45 | 13.14 |

Table 7: Evaluation results with hyper-parameter selection. The LSTM performs better than other methods in most instances.

| Method | dataset | Acc | P | R | F1 | Aff-P | Aff-R | R_A_R | R_A_P | V_ROC | V_PR |
|---|---|---|---|---|---|---|---|---|---|---|---|
| DAGMM | instance38 | 0.9768 | 0.0048 | 0.1139 | 0.0072 | 0.7389 | 0.9781 | 0.8328 | 0.0237 | 0.8081 | 0.0231 |
| | instance44 | 0.9575 | 0.0607 | 0.654 | 0.1095 | 0.6958 | 0.8153 | 0.8626 | 0.0527 | 0.8498 | 0.0505 |
| | instance15 | 0.9978 | 0.6265 | 0.8983 | 0.7386 | 0.7785 | 0.988 | 0.8982 | 0.0541 | 0.89 | 0.0554 |
| | instance23 | 0.96 | 0.0056 | 0.5938 | 0.01 | 0.6746 | 0.9365 | 0.9417 | 0.0198 | 0.9374 | 0.0199 |
| | instance14 | 0.4969 | 0.0323 | 0.9566 | 0.0624 | 0.4818 | 0.9962 | 0.5948 | 0.0269 | 0.5821 | 0.0271 |
| | instance39 | 0.9861 | 0.0649 | 0.7054 | 0.1154 | 0.5774 | 0.7579 | 0.7423 | 0.0309 | 0.7426 | 0.0297 |
| USAD | instance38 | 0.9976 | 0.078 | 0.1392 | 0.0762 | 0.6332 | 0.4486 | 0.8598 | 0.1467 | 0.8185 | 0.1275 |
| | instance44 | 0.9877 | 0.1596 | 0.4751 | 0.2342 | 0.9136 | 0.5858 | 0.9409 | 0.1776 | 0.9357 | 0.166 |
| | instance15 | 0.9892 | 0.2264 | 0.861 | 0.3526 | 0.8189 | 0.9302 | 0.992 | 0.5171 | 0.9779 | 0.4764 |
| | instance14 | 0.9956 | 0.8983 | 0.8452 | 0.8724 | 0.9452 | 0.3239 | 0.7723 | 0.1763 | 0.7161 | 0.154 |
| | instance23 | 0.9463 | 0.0057 | 0.8125 | 0.0101 | 0.6717 | 0.6 | 0.9292 | 0.2404 | 0.9263 | 0.2234 |
| | instance39 | 0.9948 | 0.1649 | 0.7054 | 0.2625 | 0.963 | 0.3333 | 0.6144 | 0.0783 | 0.6349 | 0.0827 |
| iForest | instance38 | 0.9994 | 0.6778 | 0.7722 | 0.7453 | 0.6102 | 0.7163 | 0.9385 | 0.1116 | 0.923 | 0.1082 |
| | instance44 | 0.9959 | 0 | 0 | 0 | nan | 0 | 0.7977 | 0.0369 | 0.7622 | 0.0347 |
| | instance15 | 0.9974 | 0.5894 | 0.8172 | 0.7011 | 0.8367 | 0.8832 | 0.9627 | 0.2142 | 0.9663 | 0.2055 |
| | instance14 | 0.9982 | 0.9853 | 0.9097 | 0.9473 | 0.9552 | 0.4801 | 0.9486 | 0.1809 | 0.9074 | 0.1745 |
| | instance23 | 0.9653 | 0.0065 | 0.5938 | 0.0115 | 0.7742 | 0.5933 | 0.9522 | 0.0376 | 0.9321 | 0.0379 |
| | instance39 | 0.9996 | 0.9634 | 0.7054 | 0.8352 | 0.9986 | 0.3333 | 0.8994 | 0.1219 | 0.8982 | 0.1226 |
| LSTM | instance38 | 0.9832 | 0.0488 | 0.9114 | 0.0882 | 0.5984 | 0.9988 | 0.8839 | 0.1291 | 0.8697 | 0.1047 |
| | instance44 | 0.9723 | 0.1262 | 0.9824 | 0.2196 | 0.8736 | 0.9918 | 0.9368 | 0.2201 | 0.9296 | 0.2032 |
| | instance15 | 0.9989 | 0.8885 | 0.7831 | 0.8355 | 0.9034 | 0.8604 | 0.9721 | 0.2806 | 0.9646 | 0.2731 |
| | instance14 | 0.9959 | 0.8628 | 0.9097 | 0.8867 | 0.8093 | 0.9874 | 0.9503 | 0.1775 | 0.9179 | 0.1726 |
| | instance23 | 0.9984 | 0.1699 | 0.8125 | 0.2599 | 0.7281 | 0.9464 | 0.9967 | 0.4576 | 0.9924 | 0.3873 |
| | instance39 | 0.9828 | 0.0659 | 0.9018 | 0.1177 | 0.6709 | 0.8525 | 0.9143 | 0.1524 | 0.9178 | 0.1625 |
| ATrans | instance38 | 0.9876 | 0.061 | 0.8481 | 0.1138 | 0.489 | 0.8426 | 0.6126 | 0.1524 | 0.5922 | 0.1323 |
| | instance44 | 0.989 | 0.5797 | 0.9898 | 0.7312 | 0.4965 | 0.4969 | 0.9464 | 0.7437 | 0.9066 | 0.7044 |
| | instance15 | 0.9866 | 0.1635 | 0.6814 | 0.2638 | 0.4913 | 0.9719 | 0.6662 | 0.2595 | 0.651 | 0.2442 |
| | instance23 | 0.9863 | 0 | 0 | 0 | 0.4967 | 0.9808 | 0.4991 | 0.0081 | 0.5006 | 0.0099 |
| | instance14 | 0.9878 | 0.5907 | 0.9952 | 0.7413 | 0.5109 | 0.9958 | 0.9242 | 0.7276 | 0.9231 | 0.727 |
| | instance39 | 0.9882 | 0.0788 | 0.7321 | 0.1422 | 0.5104 | 0.9879 | 0.59 | 0.1403 | 0.5892 | 0.1397 |
| DCdetector | instance38 | 0.9891 | 0.0712 | 0.8632 | 0.1368 | 0.4923 | 0.8562 | 0.649 | 0.1749 | 0.6172 | 0.1536 |
| | instance44 | 0.9902 | 0.6293 | 0.9898 | 0.7694 | 0.6016 | 0.5706 | 0.9466 | 0.768 | 0.8993 | 0.7213 |
| | instance15 | 0.9891 | 0.2236 | 0.7458 | 0.344 | 0.514 | 0.9711 | 0.6827 | 0.3056 | 0.658 | 0.2811 |
| | instance23 | 0.9901 | 0.0636 | 0.8524 | 0.1079 | 0.5123 | 0.9646 | 0.6245 | 0.1786 | 0.6034 | 0.1546 |
| | instance14 | 0.9893 | 0.6342 | 0.9469 | 0.7596 | 0.5058 | 0.9958 | 0.9179 | 0.7426 | 0.885 | 0.7102 |
| | instance39 | 0.9898 | 0.0998 | 0.8326 | 0.2043 | 0.6035 | 0.9895 | 0.6836 | 0.1834 | 0.645 | 0.2478 |

Table 8: Experimental results on part of the instances with filling mean for missing data.

| Instance | Method | Acc | P | R | F1 | A-P | A-R | R_A_R | R_A_P | V_ROC | V_PR | A_PR | A_R |
|---|---|---|---|---|---|---|---|---|---|---|---|---|---|
| instance14 | DAGMM | 44.47 | 2.94 | 95.66 | 5.7 | 48 | 99.62 | 54.15 | 2.09 | 53.1 | 2.13 | 53.67 | 46.98 |
| | USAD | 99.35 | 79.6 | 84.52 | 81.99 | 88.63 | 46.39 | 42.86 | 4.14 | 39.91 | 4.05 | 62.66 | 27.56 |
| | KNN | 24.66 | 2.19 | 96.13 | 4.28 | 51.65 | 99.65 | 81.89 | 6.99 | 78.56 | 6.8 | 82.51 | 66.67 |
| | LOF | 14.89 | 1.94 | 96.13 | 3.81 | 51.56 | 99.65 | 87.05 | 7.87 | 83.26 | 7.61 | 83.75 | 69.66 |
| | IForest | 99.69 | 91.47 | 90.97 | 91.22 | 80.46 | 87.96 | 89.25 | 14.92 | 85.44 | 14.44 | 84.86 | 71.39 |
| | COPOD | 98.24 | 0 | 0 | 0 | 95.9 | 33.3 | 94.02 | 16.66 | 90.59 | 15.91 | 91.24 | 78.69 |
| | ECOD | 98.35 | 90.48 | 6.45 | 12.04 | 96.42 | 49.97 | 86.03 | 7.77 | 82.4 | 7.41 | 83.89 | 70.07 |
| | DeepSVDD | 47.58 | 3.11 | 95.66 | 6.02 | 55.16 | 99.63 | 91.63 | 29.66 | 86.93 | 27.29 | 93.18 | 64.8 |
| | LSTM | 99.35 | 79.45 | 84.52 | 81.91 | 88.87 | 47.25 | 44.49 | 4.16 | 41.92 | 4.08 | 63.21 | 28.1 |
| | LSTM-AE | 99.34 | 79.3 | 84.52 | 81.83 | 88.75 | 47.22 | 44.13 | 4.17 | 41.82 | 4.1 | 63.43 | 28.44 |
| | LSTM-VAE | 99.34 | 79.3 | 84.52 | 81.83 | 88.75 | 47.22 | 44.13 | 4.17 | 41.82 | 4.1 | 63.43 | 28.44 |
| | Anomaly-Transformer | 98.81 | 60.28 | 95.11 | 73.8 | 51.03 | 99.52 | 90.65 | 71.65 | 89.65 | 70.67 | 80.1 | 50.73 |
| | DCdetector | 98.93 | 63.57 | 94.69 | 76.07 | 50.52 | 99.59 | 91.8 | 74.34 | 89.53 | 72.11 | 81.74 | 50.66 |
| | BeatGAN | 99.32 | 78.25 | 84.52 | 81.27 | 88.76 | 47.34 | 44.45 | 4.16 | 41.86 | 4.08 | 63.17 | 28.05 |
| instance15 | DAGMM | 99.72 | 56.87 | 89.83 | 69.65 | 76.89 | 99.3 | 89.82 | 5.41 | 89 | 5.54 | 80.57 | 81.97 |
| | USAD | 0 | 0 | 0 | 0 | 0 | 0 | 0 | 0 | 0 | 0 | 0 | 0 |
| | KNN | 97.49 | 5.8 | 40.34 | 10.14 | 79 | 82 | 74.47 | 8.38 | 70.1 | 8.19 | 42.31 | 41.29 |
| | LOF | 99.37 | 0 | 0 | 0 | 30.89 | 10.1 | 66.79 | 2.58 | 63.54 | 2.55 | 43.85 | 43.67 |
| | IForest | 98.53 | 19.16 | 99.32 | 32.13 | 80.9 | 99.08 | 96.89 | 43.58 | 97.6 | 42.15 | 97.41 | 95.7 |
| | COPOD | 99.86 | 87.5 | 68.81 | 77.04 | 96.52 | 64.82 | 98.48 | 41.36 | 98.45 | 40.54 | 97.78 | 95.84 |
| | ECOD | 99.89 | 93.16 | 73.9 | 82.42 | 94 | 74.7 | 98.75 | 38.02 | 98.54 | 35.77 | 96.83 | 96.48 |
| | DeepSVDD | 98.45 | 14.4 | 69.15 | 23.83 | 71.98 | 75.75 | 89.94 | 4.57 | 87.27 | 4.76 | 67.1 | 64.4 |
| | LSTM | 94.74 | 2.42 | 35.59 | 4.54 | 61.94 | 75.81 | 81.94 | 9.23 | 80.89 | 8.64 | 88.39 | 80.51 |
| | LSTM-AE | 94.17 | 2.18 | 35.59 | 4.11 | 71.06 | 73.19 | 81.78 | 6.39 | 80.77 | 6.34 | 88.51 | 80.87 |
| | LSTM-VAE | 94.17 | 2.18 | 35.59 | 4.11 | 71.06 | 73.19 | 81.78 | 6.39 | 80.77 | 6.34 | 88.51 | 80.87 |
| | Anomaly-Transformer | 98.66 | 16.35 | 68.14 | 26.38 | 49.13 | 97.19 | 66.62 | 25.95 | 65.1 | 24.42 | 58.12 | 50.68 |
| | DCdetector | 98.91 | 22.36 | 74.58 | 34.4 | 51.4 | 97.11 | 68.27 | 30.56 | 65.8 | 28.11 | 61.13 | 50.64 |
| | BeatGAN | 93.25 | 4.48 | 89.49 | 8.54 | 60.21 | 88.41 | 81.93 | 9.23 | 80.88 | 8.64 | 88.37 | 80.5 |
| instance23 | DAGMM | 95.36 | 0.49 | 59.38 | 0.97 | 65.48 | 96.57 | 94.17 | 1.98 | 93.74 | 1.99 | 91.72 | 89.77 |
| | USAD | 94.64 | 0.42 | 59.38 | 0.84 | 66.32 | 59.2 | 93.62 | 22.45 | 83.34 | 20.56 | 67.08 | 49.51 |
| | KNN | 55.33 | 0.09 | 100 | 0.17 | 59.44 | 100 | 99.25 | 31.13 | 97.9 | 25.37 | 91.77 | 91.11 |
| | LOF | 98.24 | 0 | 0 | 0 | 57.78 | 57.33 | 67.19 | 0.45 | 66.1 | 0.43 | 57.76 | 56.24 |
| | IForest | 95.34 | 0.48 | 59.38 | 0.96 | 69.08 | 59.33 | 95.81 | 5.17 | 94.96 | 5.31 | 92.84 | 90.37 |
| | COPOD | 99.97 | 59.38 | 59.38 | 59.38 | 99.84 | 40 | 95.44 | 23.4 | 92.94 | 21.43 | 85.41 | 81.31 |
| | ECOD | 99.97 | 67.86 | 59.38 | 63.33 | 99.85 | 40 | 95.85 | 23.63 | 93.91 | 21.67 | 85.9 | 84.47 |
| | DeepSVDD | 0.04 | 0.04 | 100 | 0.08 | 50 | 100 | 66.25 | 1.06 | 63.92 | 1.01 | 52.73 | 52.07 |
| | LSTM | 97.58 | 0.93 | 59.38 | 1.84 | 76.8 | 59.33 | 94.03 | 23.09 | 85.28 | 21.47 | 67.21 | 51.31 |
| | LSTM-AE | 97.53 | 0.91 | 59.38 | 1.8 | 77.01 | 59.35 | 94.13 | 23.55 | 85.6 | 21.14 | 66.7 | 49.55 |
| | LSTM-VAE | 97.53 | 0.91 | 59.38 | 1.8 | 77.01 | 59.35 | 94.13 | 23.55 | 85.6 | 21.14 | 66.7 | 49.55 |
| | Anomaly-Transformer | 98.63 | 0 | 0 | 0 | 49.67 | 98.08 | 49.91 | 0.81 | 50.06 | 0.99 | 49.98 | 50.72 |
| | DCdetector | 99.01 | 1.97 | 48.39 | 3.79 | 50.22 | 98.65 | 55.38 | 7.05 | 55.42 | 7.08 | 50.98 | 50.63 |
| | BeatGAN | 95.07 | 0.46 | 59.38 | 0.91 | 66.74 | 59.33 | 94.03 | 23.12 | 85.28 | 21.5 | 67.22 | 51.32 |
| instance38 | DAGMM | 83.02 | 0.06 | 11.39 | 0.13 | 61.5 | 74.65 | 72 | 0.86 | 70.72 | 0.76 | 62.42 | 57.64 |
| | USAD | 98.69 | 0.48 | 6.33 | 0.9 | 63.48 | 89.33 | 77.49 | 11.5 | 76.51 | 9.84 | 79.76 | 73.69 |
| | KNN | 72.93 | 0.32 | 93.67 | 0.65 | 56.32 | 99.99 | 89.54 | 20.09 | 89.29 | 18.1 | 92.29 | 91.32 |
| | LOF | 96.26 | 0 | 0 | 0 | 55.98 | 39.96 | 85.6 | 1.38 | 83.83 | 1.31 | 74.83 | 70.86 |
| | IForest | 99.9 | 0 | 0 | 0 | 68.3 | 29.28 | 93.45 | 8.99 | 92.45 | 8.17 | 90.9 | 91.22 |
| | COPOD | 99.9 | 40 | 7.59 | 12.77 | 40.43 | 19.68 | 95.34 | 14.72 | 94.85 | 13.26 | 91.96 | 93.94 |
| | ECOD | 99.87 | 20 | 11.39 | 14.52 | 71.33 | 83.68 | 93.85 | 20.77 | 93.09 | 18.59 | 92.08 | 93.46 |
| | DeepSVDD | 69.49 | 0.29 | 94.94 | 0.58 | 61.73 | 99.89 | 78.69 | 1.39 | 78.05 | 1.46 | 66.27 | 69.37 |
| | LSTM | 99.72 | 3.07 | 6.33 | 4.13 | 67.55 | 93.95 | 83.39 | 3.11 | 82.02 | 2.72 | 79.39 | 74.58 |
| | LSTM-AE | 99.27 | 1.1 | 7.59 | 1.92 | 60.46 | 84.28 | 79.7 | 12.39 | 79.01 | 10.99 | 83.92 | 78.9 |
| | LSTM-VAE | 99.27 | 1.1 | 7.59 | 1.92 | 60.46 | 84.28 | 79.7 | 12.39 | 79.01 | 10.99 | 83.92 | 78.9 |
| | Anomaly-Transformer | 98.8 | 0.64 | 7.59 | 1.18 | 51.04 | 84.09 | 51.04 | 2.36 | 50.9 | 2.21 | 50.28 | 50.63 |
| | DCdetector | 98.88 | 0 | 0 | 0 | 52.16 | 83.93 | 50.86 | 2.14 | 50.36 | 1.5 | 49.95 | 50.43 |
| | BeatGAN | 98.56 | 5.36 | 86.08 | 10.1 | 63.55 | 97.73 | 83.36 | 3.12 | 81.99 | 2.73 | 79.37 | 74.62 |
| instance39 | DAGMM | 98.82 | 3.87 | 33.04 | 6.92 | 64.78 | 69.75 | 73.32 | 3.31 | 73.84 | 3.17 | 83.31 | 72.58 |
| | USAD | 86.66 | 0.05 | 5.36 | 0.11 | 58.91 | 64.57 | 77.18 | 4.64 | 69.11 | 3.98 | 16 | 15.89 |
| | KNN | 26.72 | 0.18 | 100 | 0.36 | 56.63 | 100 | 59.38 | 4.09 | 61.57 | 4.29 | 91.74 | 70.32 |
| | LOF | 0.13 | 0.13 | 100 | 0.27 | 50.01 | 100 | 36.25 | 0.58 | 35.56 | 0.57 | 34.46 | 29.13 |
| | IForest | 99.95 | 87.78 | 70.54 | 78.22 | 99.69 | 33.33 | 91.29 | 12.87 | 90.82 | 12.89 | 93.6 | 86.93 |
| | COPOD | 99.89 | 81.25 | 23.21 | 36.11 | 99.98 | 22.22 | 77.68 | 10.96 | 78.38 | 11.31 | 92.1 | 81.42 |
| | ECOD | 99.88 | 66.67 | 23.21 | 34.44 | 98.6 | 33.02 | 86.23 | 19.71 | 86.19 | 19.02 | 92.4 | 85.76 |
| | DeepSVDD | 67.17 | 0.37 | 90.18 | 0.73 | 60.62 | 73.95 | 54.61 | 1.23 | 55.05 | 1.25 | 77.48 | 57.5 |
| | LSTM | 99.77 | 0 | 0 | 0 | 62.42 | 61.9 | 79.48 | 5.14 | 73.01 | 4.56 | 19.78 | 23.91 |
| | LSTM-AE | 99.69 | 13.81 | 25.89 | 18.01 | 63.12 | 59.21 | 79.5 | 5.73 | 73.41 | 5.24 | 25.68 | 28.26 |
| | LSTM-VAE | 99.69 | 13.81 | 25.89 | 18.01 | 63.12 | 59.21 | 79.5 | 5.73 | 73.41 | 5.24 | 25.68 | 28.26 |
| | Anomaly-Transformer | 98.82 | 7.88 | 73.21 | 14.22 | 51.04 | 98.79 | 59 | 14.03 | 58.92 | 13.97 | 53.92 | 50.69 |
| | DCdetector | 98.94 | 7.41 | 54.46 | 13.05 | 50.4 | 97.62 | 56.97 | 11.87 | 57.32 | 12.21 | 53.67 | 50.61 |
| | BeatGAN | 99.65 | 8.14 | 16.07 | 10.81 | 63.35 | 70.83 | 79.48 | 5.13 | 73.01 | 4.56 | 19.78 | 23.91 |
| instance44 | DAGMM | 91.47 | 15 | 99.29 | 26.06 | 66.38 | 91.57 | 78.28 | 4.31 | 78 | 4.27 | 75 | 73.68 |
| | USAD | 81.05 | 7.4 | 100 | 13.77 | 61.62 | 100 | 91.13 | 8.67 | 90.88 | 8.69 | 90.43 | 89.04 |
| | KNN | 97.23 | 35.27 | 99.45 | 52.07 | 61.48 | 97.18 | 98.99 | 54.28 | 98.57 | 52.58 | 98.09 | 92.66 |
| | LOF | 97.63 | 0 | 0 | 0 | 57.82 | 10.02 | 97.78 | 28.53 | 97.25 | 26.89 | 96.7 | 90.37 |
| | IForest | 93.7 | 19.31 | 99.53 | 32.34 | 68.95 | 86.43 | 92.27 | 8.93 | 92.18 | 9.01 | 91.49 | 90.93 |
| | COPOD | 99.92 | 95.96 | 98.98 | 97.44 | 93.49 | 33.21 | 93.23 | 10.42 | 93.13 | 10.41 | 92.59 | 91.9 |
| | ECOD | 99.94 | 96.99 | 98.98 | 97.98 | 93.54 | 33.21 | 93.39 | 10.74 | 93.3 | 10.72 | 92.63 | 92.11 |
| | DeepSVDD | 98.24 | 46.27 | 98.98 | 63.06 | 48.1 | 33.1 | 95.29 | 14.26 | 95.2 | 14.62 | 95.13 | 94.21 |
| | LSTM | 95.46 | 24.93 | 99.53 | 39.87 | 68.96 | 84.66 | 90.79 | 7.66 | 90.68 | 7.67 | 89.78 | 89.07 |
| | LSTM-AE | 88.22 | 11.38 | 99.84 | 20.42 | 65.51 | 99.59 | 91.32 | 8.28 | 91.2 | 8.33 | 90.63 | 89.43 |
| | LSTM-VAE | 88.22 | 11.38 | 99.84 | 20.42 | 65.51 | 99.59 | 91.32 | 8.28 | 91.2 | 8.33 | 90.63 | 89.43 |
| | Anomaly-Transformer | 98.93 | 58.73 | 98.98 | 73.72 | 58.86 | 56.06 | 94.54 | 74.57 | 90.44 | 70.53 | 79.36 | 50.7 |
| | DCdetector | 98.97 | 61.88 | 98.98 | 76.15 | 52.89 | 43.56 | 94.74 | 76.44 | 83.64 | 65.49 | 80.93 | 50.51 |
| | BeatGAN | 93.18 | 18.13 | 99.53 | 30.67 | 59.83 | 86.43 | 90.78 | 7.65 | 90.66 | 7.66 | 89.76 | 89.01 |

Table 9: Experimental results on part of the instances with filling zero for missing data.

| Instance | model | Acc | F1 | P | R | A-P | A-R | R_A_P | R_A_R | V_P | V_R |
|---|---|---|---|---|---|---|---|---|---|---|---|
| instance14 | USAD | 99.34 | 81.77 | 79.20 | 84.52 | 88.54 | 46.40 | 4.22 | 43.42 | 4.12 | 39.82 |
| | KNN | 1.75 | 3.45 | 1.75 | 100.00 | 50.26 | 100.00 | 6.46 | 80.87 | 6.29 | 77.59 |
| | LOF | 40.52 | 5.57 | 2.86 | 100.00 | 57.08 | 100.00 | 8.65 | 86.39 | 8.34 | 84.76 |
| | IForest | 98.23 | 16.99 | 48.10 | 10.32 | 50.21 | 33.80 | 14.54 | 92.25 | 14.41 | 92.19 |
| | COPOD | 98.24 | 0.00 | 0.00 | 0.00 | 47.66 | 16.08 | 20.40 | 95.37 | 20.24 | 95.32 |
| | ECOD | 99.70 | 90.94 | 98.42 | 84.52 | 89.27 | 45.60 | 12.24 | 92.52 | 12.15 | 92.46 |
| | DeepSVDD | 74.24 | 11.55 | 6.14 | 95.86 | 64.38 | 99.64 | 22.50 | 91.24 | 21.09 | 89.41 |
| | LSTM | 99.34 | 81.85 | 79.35 | 84.52 | 88.87 | 47.25 | 4.25 | 45.18 | 4.17 | 42.19 |
| | LSTM-AE | 99.34 | 81.72 | 79.10 | 84.52 | 88.72 | 47.22 | 4.25 | 44.76 | 4.17 | 42.03 |
| | LSTM-VAE | 99.34 | 81.72 | 79.10 | 84.52 | 88.72 | 47.22 | 4.25 | 44.76 | 4.17 | 42.03 |
| | Anomaly-Transformer | 98.78 | 74.13 | 59.07 | 99.52 | 51.09 | 99.58 | 72.76 | 92.42 | 72.70 | 92.31 |
| | DCdetector | 98.93 | 75.96 | 63.42 | 94.69 | 50.58 | 99.58 | 74.26 | 91.79 | 71.02 | 88.50 |
| | BeatGAN | 99.31 | 81.21 | 78.15 | 84.52 | 88.73 | 47.34 | 4.25 | 45.12 | 4.16 | 42.10 |
| instance15 | USAD | 4.17 | 0.73 | 0.37 | 100.00 | 50.23 | 100.00 | 5.75 | 81.18 | 5.65 | 80.01 |
| | KNN | 94.31 | 5.04 | 2.68 | 43.05 | 60.18 | 88.90 | 8.06 | 74.41 | 7.91 | 70.03 |
| | LOF | 0.35 | 0.70 | 0.35 | 100.00 | 50.33 | 100.00 | 3.75 | 65.82 | 3.79 | 62.45 |
| | IForest | 99.74 | 70.31 | 58.94 | 87.12 | 81.67 | 96.03 | 19.47 | 95.48 | 18.82 | 95.30 |
| | COPOD | 99.86 | 75.47 | 98.90 | 61.02 | 94.43 | 52.48 | 39.52 | 96.77 | 38.32 | 97.12 |
| | ECOD | 99.67 | 13.29 | 100.00 | 7.12 | 100.00 | 22.22 | 38.36 | 98.02 | 37.18 | 98.07 |
| | DeepSVDD | 99.62 | 57.90 | 47.60 | 73.90 | 75.54 | 96.72 | 25.10 | 94.95 | 22.57 | 92.99 |
| | LSTM | 94.74 | 4.54 | 2.42 | 35.59 | 61.94 | 75.81 | 9.24 | 82.52 | 8.64 | 81.50 |
| | LSTM-AE | 94.17 | 4.11 | 2.18 | 35.59 | 71.06 | 73.19 | 6.39 | 82.36 | 6.35 | 81.38 |
| | LSTM-VAE | 94.17 | 4.11 | 2.18 | 35.59 | 71.06 | 73.19 | 6.39 | 82.36 | 6.35 | 81.38 |
| | Anomaly-Transformer | 97.76 | 8.13 | 4.75 | 28.14 | 48.61 | 95.79 | 10.43 | 56.55 | 9.79 | 55.87 |
| | DCdetector | 98.91 | 34.46 | 22.40 | 74.58 | 49.49 | 97.25 | 30.59 | 68.28 | 28.18 | 65.85 |
| | BeatGAN | 93.25 | 8.54 | 4.88 | 89.49 | 60.21 | 88.41 | 9.24 | 82.45 | 8.64 | 81.45 |
| instance23 | USAD | 94.25 | 0.78 | 0.39 | 59.38 | 65.39 | 59.17 | 21.85 | 76.42 | 19.87 | 68.02 |
| | KNN | 50.36 | 0.06 | 0.03 | 40.63 | 64.11 | 79.61 | 2.01 | 66.95 | 1.66 | 64.68 |
| | LOF | 49.82 | 0.06 | 0.03 | 40.63 | 64.28 | 79.61 | 0.36 | 59.40 | 0.35 | 58.93 |
| | IForest | 99.90 | 0.00 | 0.00 | 0.00 | 86.04 | 35.45 | 0.71 | 75.36 | 0.69 | 74.26 |
| | COPOD | 99.97 | 63.33 | 67.86 | 59.38 | 99.85 | 40.00 | 34.65 | 98.68 | 29.05 | 97.84 |
| | ECOD | 99.97 | 63.33 | 67.86 | 59.38 | 99.85 | 40.00 | 23.71 | 96.49 | 21.72 | 94.05 |
| | DeepSVDD | 38.76 | 0.12 | 0.06 | 100.00 | 54.20 | 100.00 | 1.39 | 84.83 | 1.21 | 82.34 |
| | LSTM | 97.15 | 1.56 | 0.79 | 59.38 | 72.08 | 59.33 | 22.05 | 77.58 | 20.50 | 70.16 |
| | LSTM-AE | 97.25 | 1.62 | 0.82 | 59.38 | 73.27 | 59.35 | 23.02 | 77.88 | 20.54 | 70.74 |
| | LSTM-VAE | 97.25 | 1.62 | 0.82 | 59.38 | 73.27 | 59.35 | 23.02 | 77.88 | 20.54 | 70.74 |
| | Anomaly-Transformer | 98.86 | 0.00 | 0.00 | 0.00 | 51.31 | 97.85 | 0.33 | 49.62 | 0.50 | 49.77 |
| | DCdetector | 98.96 | 3.62 | 1.88 | 48.39 | 49.71 | 98.89 | 7.03 | 55.38 | 7.31 | 55.66 |
| | BeatGAN | 95.05 | 0.91 | 0.46 | 59.38 | 65.77 | 59.33 | 22.10 | 77.47 | 20.54 | 70.04 |
| instance38 | USAD | 52.48 | 0.35 | 0.18 | 88.61 | 56.43 | 99.97 | 10.99 | 77.50 | 9.28 | 76.48 |
| | KNN | 15.30 | 0.22 | 0.11 | 100.00 | 54.76 | 100.00 | 1.51 | 72.75 | 1.36 | 71.29 |
| | LOF | 0.09 | 0.19 | 0.09 | 100.00 | 50.89 | 100.00 | 1.39 | 77.19 | 1.35 | 76.02 |
| | IForest | 99.03 | 13.01 | 7.10 | 77.22 | 60.63 | 82.78 | 1.98 | 89.64 | 1.81 | 88.07 |
| | COPOD | 99.90 | 0.00 | 0.00 | 0.00 | 13.97 | 2.00 | 5.01 | 93.77 | 4.40 | 93.12 |
| | ECOD | 99.82 | 3.82 | 3.85 | 3.80 | 67.63 | 87.30 | 12.54 | 91.75 | 12.08 | 89.82 |
| | DeepSVDD | 18.51 | 0.23 | 0.12 | 100.00 | 51.77 | 100.00 | 1.94 | 76.79 | 1.92 | 76.89 |
| | LSTM | 99.78 | 5.13 | 4.31 | 6.33 | 67.87 | 93.95 | 10.52 | 80.59 | 8.39 | 79.76 |
| | LSTM-AE | 99.55 | 3.09 | 1.94 | 7.59 | 59.37 | 85.87 | 13.12 | 81.02 | 10.77 | 80.39 |
| | LSTM-VAE | 99.55 | 3.09 | 1.94 | 7.59 | 59.37 | 85.87 | 13.12 | 81.02 | 10.77 | 80.39 |
| | Anomaly-Transformer | 98.76 | 11.38 | 6.10 | 84.81 | 48.90 | 84.26 | 15.24 | 61.26 | 13.23 | 59.22 |
| | DCdetector | 98.91 | 0.00 | 0.00 | 0.00 | 49.35 | 83.86 | 1.59 | 50.54 | 1.31 | 50.30 |
| | BeatGAN | 98.65 | 10.73 | 5.72 | 86.08 | 64.80 | 98.15 | 10.53 | 80.57 | 8.40 | 79.73 |
| instance39 | USAD | 86.22 | 0.10 | 0.05 | 5.36 | 60.10 | 64.56 | 4.34 | 68.91 | 3.71 | 61.92 |
| | KNN | 35.76 | 0.39 | 0.20 | 94.64 | 56.66 | 96.09 | 2.02 | 54.33 | 2.06 | 56.10 |
| | LOF | 29.67 | 0.19 | 0.10 | 50.89 | 52.52 | 98.73 | 0.70 | 45.15 | 0.74 | 44.95 |
| | IForest | 99.87 | 0.00 | 0.00 | 0.00 | 95.53 | 10.63 | 3.05 | 67.03 | 2.99 | 68.02 |
| | COPOD | 99.87 | 35.37 | 55.77 | 25.89 | 78.04 | 27.83 | 7.13 | 86.62 | 6.72 | 86.59 |
| | ECOD | 99.88 | 35.80 | 58.00 | 25.89 | 78.09 | 27.83 | 5.97 | 83.96 | 5.62 | 83.05 |
| | DeepSVDD | 85.14 | 1.13 | 0.57 | 63.39 | 84.94 | 77.32 | 0.86 | 65.64 | 0.86 | 63.95 |
| | LSTM | 99.76 | 0.00 | 0.00 | 0.00 | 64.06 | 64.81 | 5.23 | 72.07 | 4.60 | 65.36 |
| | LSTM-AE | 99.69 | 18.41 | 14.29 | 25.89 | 63.21 | 59.21 | 5.16 | 72.35 | 4.70 | 66.07 |
| | LSTM-VAE | 99.69 | 18.41 | 14.29 | 25.89 | 63.21 | 59.21 | 5.16 | 72.35 | 4.70 | 66.07 |
| | Anomaly-Transformer | 97.97 | 5.85 | 3.12 | 47.32 | 48.61 | 96.71 | 9.40 | 56.24 | 8.47 | 55.33 |
| | DCdetector | 98.93 | 11.41 | 6.49 | 47.32 | 51.01 | 97.48 | 10.89 | 56.33 | 10.51 | 56.00 |
| | BeatGAN | 99.66 | 11.25 | 8.65 | 16.07 | 62.48 | 70.65 | 5.23 | 72.05 | 4.60 | 65.31 |
| instance44 | USAD | 79.68 | 0.15 | 0.08 | 1.02 | 54.56 | 83.33 | 2.42 | 68.04 | 2.41 | 67.76 |
| | KNN | 98.26 | 63.20 | 46.42 | 98.98 | 78.92 | 26.68 | 62.86 | 98.27 | 69.13 | 98.19 |
| | LOF | 98.25 | 63.07 | 46.28 | 98.98 | 78.74 | 26.68 | 29.08 | 97.17 | 27.89 | 96.55 |
| | IForest | 91.71 | 26.55 | 15.33 | 98.98 | 76.90 | 32.68 | 1.20 | 31.97 | 1.22 | 32.26 |
| | COPOD | 99.90 | 96.92 | 94.94 | 98.98 | 93.88 | 33.27 | 13.64 | 94.82 | 13.67 | 94.71 |
| | ECOD | 99.93 | 97.71 | 96.47 | 98.98 | 93.69 | 33.27 | 11.47 | 93.80 | 11.42 | 93.72 |
| | DeepSVDD | 99.97 | 98.98 | 98.98 | 98.98 | 67.92 | 33.10 | 35.06 | 50.71 | 43.77 | 61.55 |
| | LSTM | 93.90 | 0.27 | 0.18 | 0.55 | 62.75 | 67.99 | 2.42 | 68.16 | 2.41 | 67.87 |
| | LSTM-AE | 86.68 | 0.20 | 0.11 | 0.87 | 65.18 | 99.18 | 2.44 | 68.26 | 2.42 | 67.96 |
| | LSTM-VAE | 86.68 | 0.20 | 0.11 | 0.87 | 65.18 | 99.18 | 2.44 | 68.26 | 2.42 | 67.96 |
| | Anomaly-Transformer | 98.95 | 74.04 | 59.14 | 98.98 | 46.19 | 48.06 | 74.78 | 94.55 | 71.01 | 90.72 |
| | DCdetector | 99.00 | 76.61 | 62.49 | 98.98 | 59.10 | 56.36 | 76.58 | 94.65 | 70.38 | 88.36 |
| | BeatGAN | 91.60 | 0.20 | 0.12 | 0.55 | 52.22 | 69.76 | 2.41 | 68.10 | 2.40 | 67.81 |

Table 10: Experimental results on part of the instances with filling linear interpolation for missing data.

| Instance | model | Acc | F1 | P | R | A-P | A-R | R_A_P | R_A_R | V_P | V_R |
|---|---|---|---|---|---|---|---|---|---|---|---|
| instance14 | USAD | 99.34 | 81.77 | 79.20 | 84.52 | 88.54 | 46.40 | 5.57 | 50.08 | 5.26 | 47.33 |
| | KNN | 14.59 | 3.95 | 2.01 | 100.00 | 51.23 | 100.00 | 6.41 | 77.65 | 6.43 | 77.85 |
| | LOF | 1.75 | 3.45 | 1.75 | 100.00 | 50.26 | 100.00 | 9.11 | 84.82 | 9.01 | 84.79 |
| | IForest | 99.19 | 79.76 | 70.85 | 91.24 | 68.32 | 97.16 | 12.41 | 92.87 | 12.36 | 92.67 |
| | COPOD | 98.35 | 12.07 | 94.06 | 6.45 | 98.59 | 49.97 | 12.30 | 91.40 | 12.11 | 91.36 |
| | ECOD | 98.34 | 12.47 | 86.09 | 6.72 | 97.27 | 49.97 | 6.87 | 81.29 | 6.70 | 81.15 |
| | DeepSVDD | 1.75 | 3.45 | 1.75 | 100.00 | 50.26 | 100.00 | 15.29 | 87.50 | 14.92 | 86.75 |
| | LSTM | 99.34 | 81.85 | 79.35 | 84.52 | 88.87 | 47.25 | 5.58 | 51.32 | 5.17 | 48.74 |
| | LSTM-AE | 99.34 | 81.75 | 79.15 | 84.52 | 88.74 | 47.22 | 5.57 | 51.12 | 5.13 | 48.62 |
| | LSTM-VAE | 99.34 | 81.75 | 79.15 | 84.52 | 88.74 | 47.22 | 5.57 | 51.12 | 5.13 | 48.62 |
| | Anomaly-Transformer | 98.19 | 64.89 | 49.24 | 95.11 | 49.68 | 99.26 | 66.11 | 90.34 | 65.67 | 89.88 |
| | DCdetector | 98.91 | 75.65 | 62.99 | 94.69 | 49.57 | 99.53 | 74.05 | 91.78 | 71.93 | 89.69 |
| | BeatGAN | 99.31 | 81.21 | 78.15 | 84.52 | 88.73 | 47.34 | 5.57 | 51.27 | 5.17 | 48.67 |
| instance15 | KNN | 89.78 | 3.03 | 1.57 | 45.42 | 51.20 | 92.95 | 5.69 | 69.48 | 5.24 | 65.50 |
| | LOF | 82.22 | 0.00 | 0.00 | 0.00 | 42.43 | 17.83 | 0.78 | 53.18 | 0.77 | 50.93 |
| | IForest | 98.92 | 39.22 | 24.44 | 99.32 | 76.12 | 99.98 | 24.14 | 96.67 | 24.20 | 97.28 |
| | COPOD | 99.88 | 80.16 | 96.65 | 68.47 | 99.91 | 44.44 | 40.92 | 98.10 | 40.90 | 98.32 |
| | ECOD | 99.67 | 13.04 | 77.78 | 7.12 | 99.85 | 22.22 | 37.59 | 98.86 | 36.01 | 98.62 |
| | DeepSVDD | 99.88 | 79.53 | 94.84 | 68.47 | 89.79 | 53.71 | 37.76 | 99.04 | 37.34 | 97.88 |
| | LSTM | 94.74 | 4.54 | 2.42 | 35.59 | 61.94 | 75.81 | 9.23 | 81.94 | 8.64 | 80.90 |
| | LSTM-AE | 94.17 | 4.11 | 2.18 | 35.59 | 71.06 | 73.19 | 6.39 | 81.78 | 6.34 | 80.78 |
| | LSTM-VAE | 94.17 | 4.11 | 2.18 | 35.59 | 71.06 | 73.19 | 6.39 | 81.78 | 6.34 | 80.78 |
| | Anomaly-Transformer | 98.78 | 29.92 | 18.73 | 74.24 | 50.59 | 97.56 | 28.46 | 68.03 | 26.77 | 66.34 |
| | DCdetector | 98.91 | 34.38 | 22.34 | 74.58 | 50.59 | 97.09 | 30.51 | 68.25 | 27.52 | 65.29 |
| | BeatGAN | 93.25 | 8.54 | 4.48 | 89.49 | 60.21 | 88.41 | 9.23 | 81.93 | 8.64 | 80.89 |
| instance23 | USAD | 48.22 | 0.11 | 0.06 | 78.13 | 59.39 | 99.26 | 33.95 | 99.37 | 30.02 | 92.15 |
| | KNN | 43.45 | 0.13 | 0.07 | 100.00 | 59.00 | 100.00 | 3.06 | 78.39 | 2.60 | 80.07 |
| | LOF | 64.81 | 0.21 | 0.10 | 96.88 | 73.86 | 99.87 | 2.86 | 71.98 | 2.41 | 73.62 |
| | IForest | 85.73 | 0.53 | 0.27 | 100.00 | 61.15 | 100.00 | 3.10 | 96.33 | 2.87 | 95.05 |
| | COPOD | 99.97 | 59.38 | 59.38 | 59.38 | 99.78 | 40.00 | 42.48 | 99.57 | 37.02 | 98.33 |
| | ECOD | 99.97 | 59.38 | 59.38 | 59.38 | 99.78 | 40.00 | 44.27 | 99.63 | 38.87 | 98.89 |
| | DeepSVDD | 53.87 | 0.16 | 0.08 | 100.00 | 71.22 | 100.00 | 23.46 | 94.26 | 19.06 | 93.25 |
| | LSTM | 80.50 | 0.24 | 0.12 | 62.50 | 66.58 | 98.16 | 37.37 | 99.59 | 33.31 | 93.93 |
| | LSTM-AE | 80.07 | 0.24 | 0.12 | 62.50 | 66.83 | 98.18 | 35.38 | 99.49 | 31.51 | 93.90 |
| | LSTM-VAE | 80.07 | 0.24 | 0.12 | 62.50 | 66.83 | 98.18 | 35.38 | 99.49 | 31.51 | 93.90 |
| | Anomaly-Transformer | 98.84 | 0.00 | 0.00 | 0.00 | 50.74 | 97.92 | 0.33 | 49.61 | 0.66 | 48.97 |
| | DCdetector | 99.02 | 0.00 | 0.00 | 0.00 | 49.23 | 93.36 | 0.52 | 49.85 | 0.35 | 49.72 |
| | BeatGAN | 52.53 | 0.13 | 0.06 | 78.13 | 59.81 | 99.37 | 37.35 | 99.59 | 33.30 | 93.96 |
| instance38 | USAD | 95.78 | 0.28 | 0.14 | 5.36 | 68.08 | 85.59 | 8.30 | 70.87 | 7.00 | 69.71 |
| | KNN | 21.30 | 0.24 | 0.12 | 100.00 | 57.88 | 100.00 | 13.72 | 82.31 | 13.84 | 82.31 |
| | LOF | 22.33 | 0.24 | 0.12 | 100.00 | 57.00 | 100.00 | 2.86 | 86.18 | 2.88 | 83.52 |
| | IForest | 99.89 | 0.00 | 0.00 | 0.00 | 53.38 | 44.24 | 5.02 | 92.04 | 5.12 | 90.86 |
| | COPOD | 99.90 | 17.31 | 36.00 | 11.39 | 61.59 | 40.21 | 19.24 | 95.81 | 17.99 | 95.39 |
| | ECOD | 99.82 | 10.53 | 9.78 | 11.39 | 69.46 | 87.37 | 15.72 | 93.60 | 15.36 | 92.60 |
| | DeepSVDD | 99.55 | 4.58 | 2.87 | 11.39 | 68.91 | 99.75 | 8.62 | 78.65 | 9.05 | 77.15 |
| | LSTM | 99.69 | 2.26 | 1.60 | 3.80 | 67.92 | 93.85 | 2.43 | 74.49 | 2.15 | 73.06 |
| | LSTM-AE | 97.79 | 6.73 | 3.50 | 84.81 | 58.62 | 82.66 | 11.02 | 71.61 | 9.74 | 70.47 |
| | LSTM-VAE | 97.79 | 6.73 | 3.50 | 84.81 | 58.62 | 82.66 | 11.02 | 71.61 | 9.74 | 70.47 |
| | Anomaly-Transformer | 98.00 | 0.00 | 0.00 | 0.00 | 49.46 | 84.06 | 1.25 | 50.25 | 1.06 | 50.08 |
| | DCdetector | 98.90 | 0.00 | 0.00 | 0.00 | 50.71 | 84.09 | 1.74 | 50.66 | 1.34 | 50.31 |
| | BeatGAN | 96.83 | 4.87 | 2.50 | 86.08 | 61.83 | 94.09 | 2.44 | 74.56 | 2.16 | 73.14 |
| instance39 | USAD | 85.64 | 0.10 | 0.05 | 5.36 | 58.99 | 64.60 | 4.46 | 67.89 | 3.81 | 60.20 |
| | KNN | 20.65 | 0.34 | 0.17 | 100.00 | 54.25 | 100.00 | 3.87 | 63.48 | 3.96 | 65.24 |
| | LOF | 23.77 | 0.34 | 0.17 | 98.21 | 53.38 | 99.90 | 1.44 | 50.11 | 1.52 | 50.72 |
| | IForest | 99.89 | 27.48 | 94.74 | 16.07 | 99.93 | 22.21 | 15.51 | 82.18 | 15.15 | 82.10 |
| | COPOD | 99.89 | 27.48 | 94.74 | 16.07 | 99.93 | 22.21 | 17.29 | 82.42 | 16.98 | 82.53 |
| | ECOD | 99.87 | 25.00 | 56.25 | 16.07 | 81.24 | 36.33 | 31.29 | 89.30 | 29.04 | 88.96 |
| | DeepSVDD | 40.10 | 0.42 | 0.21 | 94.64 | 67.06 | 80.96 | 4.43 | 51.22 | 4.47 | 51.31 |
| | LSTM | 99.76 | 0.00 | 0.00 | 0.00 | 62.43 | 61.90 | 4.91 | 70.60 | 4.39 | 63.53 |
| | LSTM-AE | 99.69 | 18.12 | 13.94 | 25.89 | 60.77 | 54.08 | 5.61 | 70.64 | 5.12 | 64.18 |
| | LSTM-VAE | 99.69 | 18.12 | 13.94 | 25.89 | 60.77 | 54.08 | 5.61 | 70.64 | 5.12 | 64.18 |
| | Anomaly-Transformer | 98.66 | 8.62 | 4.74 | 47.32 | 47.10 | 96.61 | 9.10 | 55.41 | 9.42 | 55.71 |
| | DCdetector | 98.91 | 12.71 | 7.19 | 54.46 | 50.04 | 97.79 | 11.62 | 56.78 | 11.82 | 57.05 |
| | BeatGAN | 99.63 | 0.00 | 0.00 | 0.00 | 62.80 | 70.40 | 4.91 | 70.54 | 4.39 | 63.49 |
| instance44 | USAD | 24.66 | 3.86 | 1.97 | 100.00 | 55.01 | 100.00 | 1.55 | 32.32 | 1.52 | 32.54 |
| | KNN | 37.28 | 4.58 | 2.34 | 99.45 | 53.16 | 97.18 | 3.14 | 39.99 | 3.60 | 40.40 |
| | LOF | 98.49 | 0.00 | 0.00 | 0.00 | 0.00 | 0.00 | 2.61 | 41.98 | 3.00 | 43.06 |
| | IForest | 77.86 | 11.94 | 6.35 | 99.13 | 48.02 | 81.26 | 2.58 | 62.72 | 2.57 | 61.92 |
| | COPOD | 99.97 | 98.86 | 98.74 | 98.98 | 93.15 | 32.31 | 9.38 | 90.52 | 9.59 | 90.38 |
| | ECOD | 99.95 | 98.51 | 98.05 | 98.98 | 92.80 | 32.31 | 7.07 | 88.31 | 7.03 | 88.20 |
| | DeepSVDD | 1.51 | 2.98 | 1.51 | 100.00 | 50.18 | 100.00 | 10.32 | 88.98 | 9.99 | 88.64 |
| | LSTM | 35.07 | 4.43 | 2.27 | 99.53 | 60.82 | 84.66 | 1.35 | 34.01 | 1.42 | 34.43 |
| | LSTM-AE | 27.55 | 4.00 | 2.04 | 99.69 | 57.06 | 98.70 | 1.44 | 32.68 | 1.41 | 32.97 |
| | LSTM-VAE | 27.55 | 4.00 | 2.04 | 99.69 | 57.06 | 98.70 | 1.44 | 32.68 | 1.41 | 32.97 |
| | Anomaly-Transformer | 98.90 | 73.12 | 57.97 | 98.98 | 49.65 | 49.69 | 74.37 | 94.64 | 70.44 | 90.66 |
| | DCdetector | 99.02 | 76.94 | 62.93 | 98.98 | 60.16 | 57.06 | 76.80 | 94.66 | 72.13 | 89.93 |
| | BeatGAN | 32.31 | 4.26 | 2.17 | 99.53 | 51.67 | 86.43 | 1.35 | 34.06 | 1.43 | 34.48 |

