# OpenReview forum: "Benchmarking Multivariate Time Series Anomaly Detection with Large-Scale Real-World Datasets"
_ICLR.cc/2024/Conference — ICLR 2024 Conference Withdrawn Submission_

### Official Review · Reviewer_HrBh · 2023-10-31

**Soundness:** 2 fair
**Presentation:** 2 fair
**Contribution:** 2 fair
**Rating:** 3
**Confidence:** 5

**Summary:**

The paper addresses the challenges of evaluating multivariate time series anomaly detection (MTSAD) methods, such as the lack of public datasets, the diversity of metrics, and the fair comparison of models.It proposes a benchmark for MTSAD, with a new large-scale real-world dataset collected from an AIOps system for a real-time data warehouse, a review and comparison of popular evaluation metrics, and  experiments of 14 methods on various datasets.It also acknowledges the potential of deep learning in MTSAD but emphasizes the need for more work in real-world applications.It highlights the limitations of existing benchmarks and metrics, emphasizing the need for further research in this area.

**Strengths:**

●	The paper addresses the limitations of existing benchmarks for multivariate time series anomaly detection (MTSAD) and proposes a new benchmark framework.
●	It presents a novel benchmark for MTSAD, with the largest real-world dataset to date, which covers diverse anomaly types and scenarios in a complex and large-scale system.
●	The paper compares 14 machine learning and deep learning models using various evaluation metrics, providing an analysis of their performance.
●	It makes the benchmark and dataset open-sourced, which can facilitate future research and applications in MTSAD.

**Weaknesses:**

●	It is unclear how the new benchmark addresses the flaws in existing benchmarks and datasets, as mentioned in the paper.
●	The paper does not discuss the limitations or potential biases of the proposed benchmark and dataset.
●	It does not provide a thorough analysis of the strengths and weaknesses of the compared algorithms, focusing more on the experimental results.
Ever since the benchmark paper in Neurips22, I see a good number of similar papers. Benchmarking is an exercise I must applaud but detecting anomalies must be done on data sets which have a wide range of anomalies (1- many). The methods which are evaluated--how do they perform on test data sets with no anomalies? How many false flags do they raise?

**Questions:**

●	How does the proposed benchmark address the limitations and flaws of existing benchmarks and datasets?
●	What are the potential limitations or biases of the new benchmark and dataset, and how are they addressed?
●	Can the authors provide a more in-depth analysis of the strengths and weaknesses of the compared algorithms, beyond the experimental results?

Have you tested the algorithms on public datasets (AWS, yahoo etc)?

---

### Official Review · Reviewer_NB3i · 2023-10-31

**Soundness:** 2 fair
**Presentation:** 3 good
**Contribution:** 2 fair
**Rating:** 3
**Confidence:** 4

**Summary:**

The paper introduces a comprehensive benchmark study on multivariate time series anomaly detection (MTSAD) with a new large-scale AIOps dataset.
The authors evaluated various traditional machine learning-based and advanced deep learning-based anomaly detection methods and employed ten evaluation metrics to assess the performance of these methods.
The experiments were conducted on diverse datasets from various domains to ensure the generalizability of the findings.

**Strengths:**

S1. **Diverse Dataset Evaluation:**
The paper uses a wide range of public MTS datasets and their collected AIOps datasets that cover multiple domains, which enhances the generalizability of the study.

S2. **Comprehensive Analysis:**
The authors provide a thorough analysis of different MTSAD methods, both traditional and state-of-the-art. This breadth is beneficial for understanding the landscape of MTSAD.

S3. **Practical Impacts:**
This benchmark study provides many key insights for academic researchers and industry experts by highlighting the best-performing methods across different metrics on different data domains.

**Weaknesses:**

W1. **Efficiency and Scalability Concerns:**
As claimed by the authors, "the accuracy and efficiency of the MTTD can have a significant impact on the overall performance and reliability of the systems.'' While this paper provides a comprehensive analysis of various MTSAD methods in terms of accuracy metrics, it would significantly benefit from an evaluation of efficiency and scalability aspects. This is crucial given the large-scale nature of the datasets and the potential real-world deployment of these methods.

W2. **Robustness of Methods:**
In addition to the primary evaluation metrics, it is essential to consider the robustness of the MTS anomaly detection methods. Variability in results due to factors such as random seed settings for methods like IForest or initial weight configurations for deep learning methods can significantly impact the reliability of the method in real-world scenarios.

W3. **Statistical Test**:
For a benchmark endeavor, a comprehensive statistical test might be necessary. The authors could add a statistical test on multiple methods based on all datasets used for each metric. This will provide a clear yet insightful view of the rank of different methods and differences among various metrics. For example, the SMD can be divided into 28 subsets, so the authors could treat them as separate datasets, just as treating different instances in AIOps as different datasets.

W4. **Inclusion of Advanced Methods:**
While the paper provides an extensive list of methods evaluated, it seems to miss out on some of the advanced methods. For example, they have experimented with the PSM dataset but appear to have omitted the RANSynCoders method, which is intrinsically linked with this dataset. RANSynCoders, given its relevance to the PSM dataset, could provide insightful results and comparisons.

* Abdulaal, Ahmed, Zhuanghua Liu, and Tomer Lancewicki. "Practical approach to asynchronous multivariate time series anomaly detection and localization." In Proceedings of the 27th ACM SIGKDD conference on knowledge discovery & data mining, pp. 2485-2494. 2021.

W5. **Parameter Settings:**
The paper might benefit from a clearer explanation of certain methodological choices, such as parameter settings for various algorithms.

W6. **Justification of  AIOps Datasets:**
The authors have rigorously introduced the limitations of the existing datasets and proposed a new large-scale dataset along with its background and collection mechanism. However, they could provide more justifications (e.g., using some case study) on why such datasets are able to alleviate the aforementioned issues in the existing ones.

**Questions:**

Regarding W1:

Q1. For each method, can the authors report the time taken for training/fitting and the time taken for testing? This would not only give insights into the efficiency of each method but also highlight any potential trade-offs between accuracy and efficiency.

Q2. Can the authors analyze how the running time of each method changes with respect to the number of dimensions (#dims) and the number of timestamps (#timestamps)? This can reveal potential bottlenecks or inefficiencies in certain methods and provide guidance on which methods are more suitable for specific types of datasets.

Q3. Given the emphasis on large-scale real-world datasets, can they show how each method scales with increasing data size? This will provide crucial insights for practitioners considering deploying these methods in real-world scenarios where data scales can be massive.

Regarding W2:

Q4: To ensure a comprehensive evaluation of robustness, it would be beneficial if the authors conducted multiple runs of each method with varied configurations (e.g., different random seeds and initial weight settings). Then, the authors could report the standard deviations of the results for each method. This would provide insights into the variability or consistency of the method's performance across different runs or configurations.

Regarding W4:

Q5: Can the authors consider adding RANSynCoders to their benchmarking methods, especially when using the PSM dataset? This will not only enhance the paper's credibility but also provide a more holistic view of how different methods perform on this specific dataset.

**Details Of Ethics Concerns:**

No.

---

### Official Review · Reviewer_Gnj7 · 2023-11-01

**Soundness:** 2 fair
**Presentation:** 3 good
**Contribution:** 2 fair
**Rating:** 3
**Confidence:** 4

**Summary:**

The paper designs a new benchmarking suite for multivariate time-series anomaly detection. In particular, the work provides a very large datasets for this task coming from AIOps monitoring application. In addition, the work combines this dataset with prior multivariate datasets and performs an experimental evaluation assessing evaluation measures as well.

**Strengths:**

- Benchmarking anomaly detection methods especially for multivariate time series data is critical, challenging, and timely problem
- New datasets are necessary in the area

**Weaknesses:**

- Technical novelty is limited due to the nature of this paper
- Experimental settings for detectors are not clear
- Datasets already used in prior works
- Results on the new datasets are somewhat "disappointing."

**Questions:**

- Technical novelty is limited due to the nature of this paper

Due to the nature of this paper (benchmarking) the technical depth is somewhat low. Benchmark tracks in sister conferences might be more suitable. No matter what, apart from sharing a new dataset, the remaining components/datasets of the benchmark already exist

- Experimental settings for detectors are not clear

The settings for the detectors are not clear. Tons of parameters necessary to be tuned, unclear how it is happening, if baselines achieve their best performance etc.

- Datasets already used in prior works

Prior benchmarks (Paparizzos et al) have used these multivariate datasets and converted them to univariate. The eval seems new but the community seems to be well aware of these multivariate datasets

- Results on the new datasets are somewhat "disappointing."

For the new datasets, most methods perform the same (see R_A_P results in table 4 and similar metrics, many methods have more or less similar eval values). We haven't seen a single plot of the new dataset, to understand how easy/difficult it is to detect anomalies. Other methods show huge differences (e.g., going from 15 to 88.. this is an indication of badly tuned parameters etc. it's very difficult to believe such huge jumps in accuracy. In other measures)

---

### Official Review · Reviewer_61C3 · 2023-11-03

**Soundness:** 3 good
**Presentation:** 4 excellent
**Contribution:** 3 good
**Rating:** 6
**Confidence:** 4

**Summary:**

The paper identifies large real-world datasets to use as benchmarks for multivariate time series anomaly detection (MTSAD). The paper also identifies various metrics that can be used to evaluate MTSAD algorithms and compare their performances. The paper discusses the benefits and drawbacks of the various metrics. Finally, the paper compares various algorithms on the benchmark datasets using the identified metrics, with particular focus on comparing deep learning based methods with classical methods.

**Strengths:**

1. The paper seems fairly comprehensive in the set of metrics that it uses.
2. The paper lays out the benchmark framework quite well in figure 1 and the text explanation.
3. The comparisons are quite comprehensive between the main text and the appendix and all the tables and figures.

**Weaknesses:**

1. More detailed explanations of the datasets chosen and why they were chosen are needed. The datasets should cover a wide range of characteristics, such as numbers and types of variables, how redundant the input variables are, how variable the processes are, etc. This should be discussed.
2. The importance of each metric for each benchmark dataset should also be discussed. In the main text, the most important metric for each dataset should be highlighted with a brief explanation for why it is important should be given. More details on this can then be given in the appendix.

**Questions:**

Minor questions:
1. What are SLAs? This abbreviation has not been defined in the paper.